# A translation proofreader of archaeal origin imparts multi-aldehyde stress tolerance to land plants

Pradeep Kumar[1,2,3], Ankit Roy[1], Shivapura Jagadeesha Mukul[1,2,3], Avinash Kumar Singh[1], Dipesh Kumar Singh[1], Aswan Nalli[1], Pujaita Banerjee[1], Kandhalu Sagadevan Dinesh Babu[1], Bakthisaran Raman[1], Shobha P Kruparani[1], Imran Siddiqi[1,2], Rajan Sankaranarayanan[1,2,3]*

[1]CSIR–Centre for Cellular and Molecular Biology, Hyderabad, India; [2]Academy of Scientific and Innovative Research (AcSIR), CSIR–CCMB Campus, Hyderabad, India; [3]Academy of Scientific and Innovative Research (AcSIR), Ghaziabad, India

*For correspondence:
sankar@ccmb.res.in

**Abstract** Aldehydes, being an integral part of carbon metabolism, energy generation, and signalling pathways, are ingrained in plant physiology. Land plants have developed intricate metabolic pathways which involve production of reactive aldehydes and its detoxification to survive harsh terrestrial environments. Here, we show that physiologically produced aldehydes, i.e., formaldehyde and methylglyoxal in addition to acetaldehyde, generate adducts with aminoacyl-tRNAs, a substrate for protein synthesis. Plants are unique in possessing two distinct chiral proofreading systems, D-aminoacyl-tRNA deacylase1 (DTD1) and DTD2, of bacterial and archaeal origins, respectively. Extensive biochemical analysis revealed that only archaeal DTD2 can remove the stable D-aminoacyl adducts on tRNA thereby shielding archaea and plants from these system-generated aldehydes. Using *Arabidopsis* as a model system, we have shown that the loss of DTD2 gene renders plants susceptible to these toxic aldehydes as they generate stable alkyl modification on D-aminoacyl-tRNAs, which are recycled only by DTD2. Bioinformatic analysis identifies the expansion of aldehyde metabolising repertoire in land plant ancestors which strongly correlates with the recruitment of archaeal DTD2. Finally, we demonstrate that the overexpression of DTD2 offers better protection against aldehydes than in wild type *Arabidopsis* highlighting its role as a multi-aldehyde detoxifier that can be explored as a transgenic crop development strategy.

## eLife assessment

The work is a **fundamental** contribution towards understanding the role of archaeal and plant D-aminoacyl-tRNA deacylase 2 (DTD2) in deacylation and detoxification of D-Tyr-tRNATyr modified by various aldehydes produced as metabolic byproducts in plants. It integrates **convincing** results from both in vitro and in vivo experiments to address the long-standing puzzle of why plants outperform bacteria in handling reactive aldehydes and suggests a new strategy for stress-tolerant crops. A limitation of the study is the lack of evidence for accumulation of toxic D-aminoacyl tRNAs and impairment of translation in plant cells lacking DTD2.

## Introduction

Reactive metabolites are an integral part of biological systems as they fuel a plethora of fundamental processes of life. Metabolically generated aldehydes are chemically diverse reactive metabolites such as formaldehyde (1-C), acetaldehyde (2-C), and methylglyoxal (MG; 3-C). Formaldehyde integrates

**Figure 1.** Aldehydes generate N-alkylated-aa-tRNA adducts. Thin-layer chromatography (TLC) showing modification on L- and D-Tyr-tRNA^Tyr by (**A**) formaldehyde, acetaldehyde, propionaldehyde, butyraldehyde, valeraldehyde, isovaleraldehyde, decanal and (**B**) MG (AMP: adenine monophosphate which corresponds to free tRNA, whereas Tyr-AMP and modified-Tyr-AMP correspond to unmodified and modified Tyr-tRNA^Tyr). These modifications were generated by incubating 2 µM aa-tRNA with 100 mM of respective aldehydes along with 20 mM sodium cyanoborohydride (in 100 mM potassium acetate [pH 5.4]) as a reducing agent at 37°C for 30 min. Mass spectra showing (**C**) D-Phe-tRNA^Phe, (**D**) formaldehyde-modified D-Phe-tRNA^Phe, (**E**) propionaldehyde-modified D-Phe-tRNA^Phe, (**F**) butyraldehyde-modified D-Phe-tRNA^Phe, (**G**) MG-modified D-Phe-tRNA^Phe. (**H**) Graph showing the effect of increasing chain length of aldehyde on modification propensity with aa-tRNA at two different concentrations of various aldehydes (n=3). Effect of (**I**) formaldehyde and (**J**) MG modification on stability of ester linkage in D-aminoacyl-tRNA (D-aa-tRNA) under alkaline conditions (n=3).

The online version of this article includes the following source data and figure supplement(s) for figure 1:

*Figure 1 continued on next page*

*Figure 1 continued*

**Source data 1.** Biochemical data for the modification susceptibility of L-Ala-tRNA$^{Ala}$ by multiple aldehydes and stability of formaldehyde- and MG-modified and unmodified D-Tyr-tRNA$^{Tyr}$ substrates under alkaline conditions.

**Source data 2.** Table showing the expected and observed mass change upon aldehyde modification on D-Phe-tRNA$^{Phe}$ by electrospray ionisation mass spectrometry (ESI-MS)/MS.

**Figure supplement 1.** Aldehydes modify the amino group of amino acids in D-aminoacyl-tRNAs (D-aa-tRNAs).

various carbon metabolic pathways and is produced as a by-product of oxidative demethylation by various enzymes (*Jardine et al., 2017*; *Song et al., 2013*; *Trézl et al., 1998*; *Loenarz and Schofield, 2008*; *Shi et al., 2004*; *Walport et al., 2012*) whereas acetaldehyde is an intermediate of anaerobic fermentation (*Tadege and Kuhlemeier, 1997*). Alternatively, MG is produced via the glycolysis pathway from dihydroxyacetone phosphate and glyceraldehyde-3-phosphate, oxidative deamination of glycine and threonine, fatty acid degradation, and auto-oxidation of glucose inside the cell (*Mostofa et al., 2018*). These aldehydes are involved in carbon metabolism (*Jardine et al., 2017*; *Song et al., 2013*; *Trézl et al., 1998*; *Burgos-Barragan et al., 2017*; *Hill et al., 2011*), energy generation (*Tadege and Kuhlemeier, 1997*), and signalling (*Mostofa et al., 2018*; *Kosmachevskaya et al., 2017*), respectively, in all domains of life. In addition to the three aldehydes discussed above, plants also produce a wide range of other aldehydes under various biotic and abiotic stresses (*Mostofa et al., 2018*; *Jardine et al., 2009*). Despite their physiological importance, these aldehydes become genotoxic and cellular hazards at higher concentrations as they irreversibly modify the free amino group of various essential biological macromolecules like nucleic acids, proteins, lipids, and amino acids (*Seitz and Stickel, 2007*; *Fang and Vaca, 1997*; *Matsuda et al., 1999*; *Fang and Vaca, 1995*; *Carlsson et al., 2014*). Increased levels of formaldehyde and MG lead to toxicity in various life forms like bacteria (*Chen et al., 2016*) and mammals (*Burgos-Barragan et al., 2017*; *Pontel et al., 2015*; *Allaman et al., 2015*). However, archaea and plants possess these aldehydes in high amounts (>25-fold) (*Figure 1—figure supplement 1A*), yet there is no evidence of toxicity (*Trézl et al., 1998*; *Miller et al., 2017*; *Dingler et al., 2020*; *Li et al., 2017*; *Kimmerer and Macdonald, 1987*; *Quintanilla et al., 2007*; *Yadav et al., 2005*; *Rabbani and Thornalley, 2014*; *Wang et al., 2019*; *Baskaran et al., 1989*). This suggests that both archaea and plants have evolved specialised protective mechanisms against toxic aldehyde flux.

Using genetic screening Takashi et al. have identified a gene, called GEK1 at that time, essential for the protection of plants from ethanol and acetaldehyde (*Fujishige et al., 2004*; *Hirayama et al., 2004*). Later, using biochemical and bioinformatic analysis, GEK1 was identified to be a homolog of archaeal D-aminoacyl-tRNA deacylase (DTD) (*Wydau et al., 2007*). DTDs are *trans* acting, chiral proofreading enzymes involved in translation quality control and remove D-amino acids mischarged onto tRNAs (*Calendar and Berg, 1967*; *Soutourina et al., 1999*; *Soutourina et al., 2000*; *Kuncha et al., 2019*; *Kumar et al., 2022*). DTD function is conserved across all life forms where DTD1 is present in bacteria and eukaryotes, DTD2 in land plants and archaea (*Wydau et al., 2007*; *Ferri-Fioni et al., 2006*), and DTD3 in cyanobacteria (*Wydau et al., 2009*). All DTDs are shown to be important in protecting organisms from D-amino acids (*Wydau et al., 2007*; *Calendar and Berg, 1967*; *Soutourina et al., 2000*; *Ferri-Fioni et al., 2006*; *Wydau et al., 2009*). In addition, DTD2 was also found to be involved in protecting plants against ethanol and acetaldehyde (*Fujishige et al., 2004*; *Hirayama et al., 2004*; *Wydau et al., 2007*). Recently, we identified the biochemical role of archaea-derived DTD2 gene in alleviating acetaldehyde stress in addition to resolving organellar incompatibility of bacteria-derived DTD1 in land plants (*Mazeed et al., 2021*; *Kumar et al., 2023*). We have shown that acetaldehyde irreversibly modifies D-aminoacyl-tRNAs (D-aa-tRNA) and only DTD2 can recycle the modified D-aa-tRNAs thus replenishing the free tRNA pool for further translation (*Mazeed et al., 2021*). Like acetaldehyde, elevated aldehyde spectrum (*Figure 1—figure supplement 1A*) in plants and archaea pose a threat to the translation machinery. The unique presence of DTD2 in organisms with elevated aldehyde spectrum (plants and archaea) and its indispensable role in acetaldehyde tolerance prompted us to investigate the role of archaeal DTD2 in safeguarding translation apparatus of plants from various physiologically abundant toxic aldehydes.

Here, our in vivo and biochemical results suggest that formaldehyde and MG lead to toxicity in DTD2 mutant plants through D-aa-tRNA modification. Remarkably, out of all the aldehyde-modified D-aa-tRNAs tested, only the physiologically abundant ones (i.e. D-aa-tRNAs modified by formaldehyde or

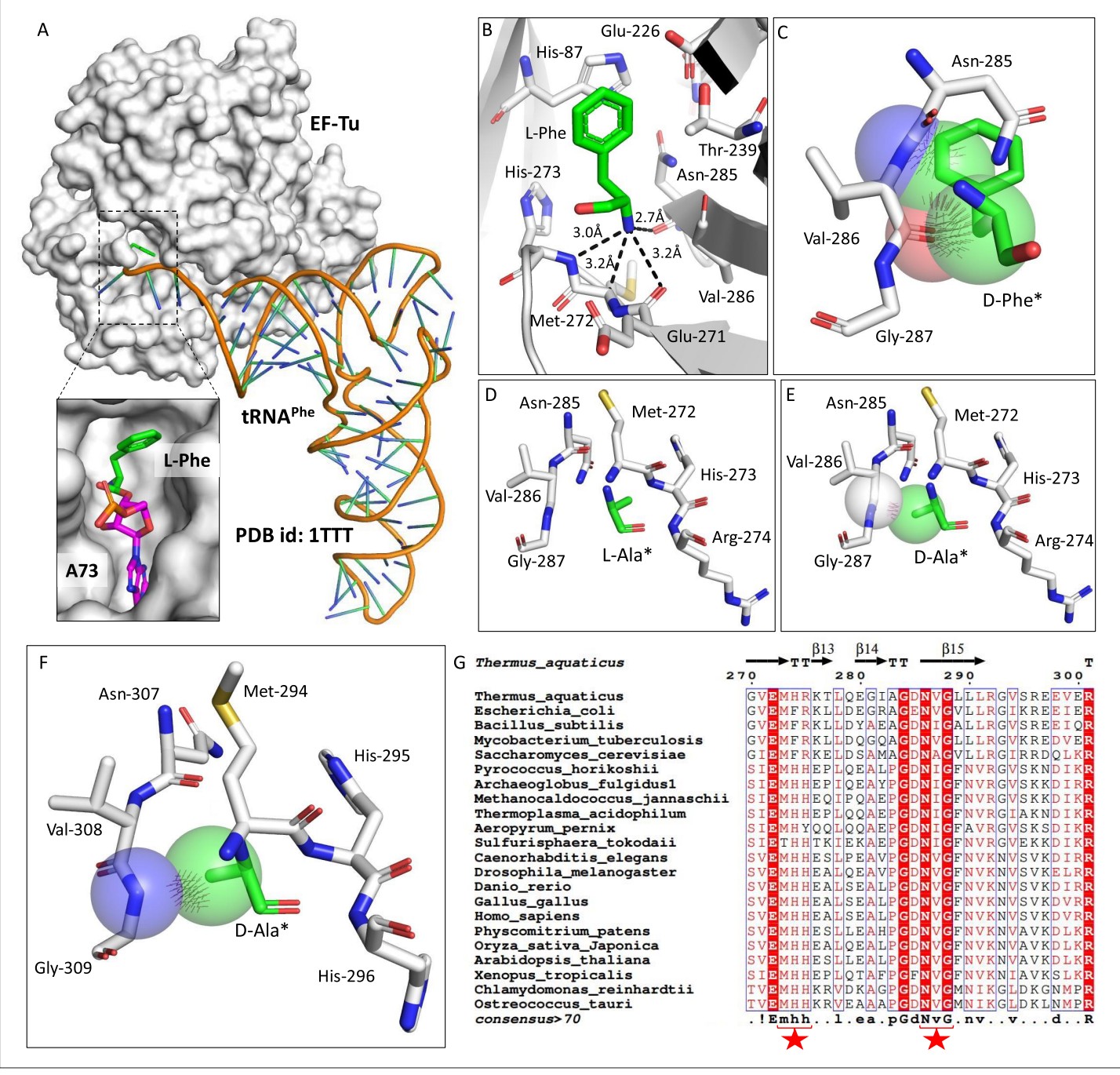

**Figure 2.** Elongation factor enantioselects aa-tRNAs through D-chiral rejection mechanism. (**A**) Surface representation showing the cocrystal structure of elongation factor thermo unstable (EF-Tu) with L-Phe-tRNAPhe. Zoomed-in image showing the binding of L-phenylalanine with side chain projected outside of binding site of EF-Tu (PDB id: 1TTT). (**B**) Zoomed-in image of amino acid binding site of EF-Tu bound with L-phenylalanine showing the selection of amino group of amino acid through main chain atoms (PDB id: 1TTT). (**C**) Modelling of D-phenylalanine in the amino acid binding site of EF-Tu shows severe clashes with main chain atoms of EF-Tu. Modelling of smallest chiral amino acid, alanine, in the amino acid binding site of EF-Tu shows (**D**) no clashes with L-alanine and (**E**) clashes with D-alanine. (**F**) Modelling of D-alanine in the amino acid binding site of eEF-1A shows clashes with main chain atoms. (*Represents modelled molecule.) (**G**) Structure-based sequence alignment of elongation factor from bacteria, archaea, and eukaryotes (both plants and animals) showing conserved amino acid binding site residues. (Key residues are marked with red star.)

The online version of this article includes the following figure supplement(s) for figure 2:

**Figure supplement 1.** Elongation factor protects L-aa-tRNAs from aldehyde modification.

methylglyoxal) were deacylated by both archaeal and plant DTD2s. Therefore, plants have recruited archaeal DTD2 as a potential detoxifier of all toxic aldehydes rather than only acetaldehyde as earlier envisaged. Furthermore, DTD2 overexpressing *Arabidopsis* transgenic plants demonstrate enhanced multi-aldehyde resistance that can be explored as a strategy for crop improvement.

## Results

### Aldehydes modify D-aa-tRNAs to disrupt protein synthesis

The presence of large amounts of chemically diverse aldehydes in plants and archaea (*Figure 1—figure supplement 1A*) encouraged us to investigate their influence on aa-tRNAs, a key component of the translational machinery. We incubated aa-tRNAs with diverse aldehydes (from formaldehyde [1-C] to decanal [10-C] including MG [3-C]) and investigated adduct formation with thin-layer chromatography (TLC) and electrospray ionisation mass spectrometry (ESI-MS). We observed that aldehydes modified aa-tRNAs irrespective of amino acid chirality (*Figure 1A–G* and *Figure 1—figure supplement 1B*). The mass change from formaldehyde, propionaldehyde, butyraldehyde, and MG modification corresponds to a methyl, propyl, butyl, and acetonyl group, respectively (*Figure 1C–G*). Tandem fragmentation (MS$^2$) of aldehyde-modified D-aa-tRNAs showed that all the aldehydes selectively modify only the amino group of amino acids in D-aa-tRNAs (*Figure 1—figure supplement 1C–G*). Interestingly, upon a comparison of modification strength, the propensity of modification decreased with increase in the aldehyde chain length with no detectable modification on decanal-treated aa-tRNAs (*Figure 1A and H*). The chemical reactivity of aldehyde is dictated by its electrophilicity (*LoPachin and Gavin, 2014*). The electrophilicity of saturated aldehydes decreases with the increasing chain length of aldehyde (*LoPachin and Gavin, 2014*; *Pratihar, 2014*), thereby reducing modification propensity. Exceptionally, the modification propensity of MG is much higher than propionaldehyde (*Figure 1H*) which is also a three-carbon system (*Figure 1—figure supplement 1H*) and it is likely due to the high electrophilicity of the carbonyl carbon (*LoPachin and Gavin, 2014*). Also, the aldehydes with higher propensity of modification are present in higher amounts in plants and archaea (*Figure 1—figure supplement 1A*). Further, we investigated the effect of aldehyde modification on the stability of ester linkage of aa-tRNAs by treating them with alkaline conditions. Strikingly, even the smallest aldehyde modification stabilised the ester linkage by ~13-fold when compared with unmodified aa-tRNA (*Figure 1I–J*).

Elongation factor thermo unstable (EF-Tu) is shown to protect L-aa-tRNAs from acetaldehyde modification (*Mazeed et al., 2021*). EF-Tu-based protection of L-aa-tRNAs can be extended to any aldehydes with similar or bigger size than acetaldehyde but not formaldehyde. We sought to investigate the elongation factor-based protection against formaldehyde. To understand this, we have done a thorough sequence and structural analysis. We analysed the aa-tRNA-bound elongation factor structure from bacteria (PDB ids: 1TTT) and found that the side chain of amino acid in the amino acid binding site of EF-Tu is projected outside (*Figure 2A* and *Figure 2—figure supplement 1A*). In addition, the amino group of amino acid is tightly selected by the main chain atoms of elongation factor thereby lacking a space for aldehydes to enter and then modify the L-aa-tRNAs and Gly-tRNAs (*Figure 2B* and *Figure 2—figure supplement 1B*). Modelling of D-amino acid (either D-phenylalanine or smallest chiral amino acid, D-alanine) in the same site shows serious clashes with main chain atoms of EF-Tu, indicating a D-chiral rejection during aa-tRNA binding by elongation factor (*Figure 2C–E*). Next, we superimposed the tRNA-bound mammalian (from *Oryctolagus cuniculus*) eEF-1A cryoEM structure (PDB id: 5LZS) with bacterial structure to understand the structural differences in terms of tRNA binding and found that elongation factor binds tRNA in a similar way (*Figure 2—figure supplement 1C–D*). Modelling of D-alanine in the amino acid binding site of eEF-1A also shows serious clashes with main chain atoms, indicating a general theme of D-chiral rejection during aa-tRNA binding by elongation factor (*Figure 2F* and *Figure 2—figure supplement 1E*). Structure-based sequence alignment of elongation factor from bacteria, archaea, and eukaryotes (both plants and mammals) shows a strict conservation of amino acid binding site (*Figure 2G*). Minor differences near the amino acid side chain binding site (as indicated in Wolfson and Knight, *FEBS Letters*, 2005) might induce the amino acid specific binding differences, if any (*Figure 2—figure supplement 1F*). However, those changes will have no influence when the D-chiral amino acid enters the pocket, as the whole side chain would clash with the active site. To confirm these structural and sequence analyses biochemically,

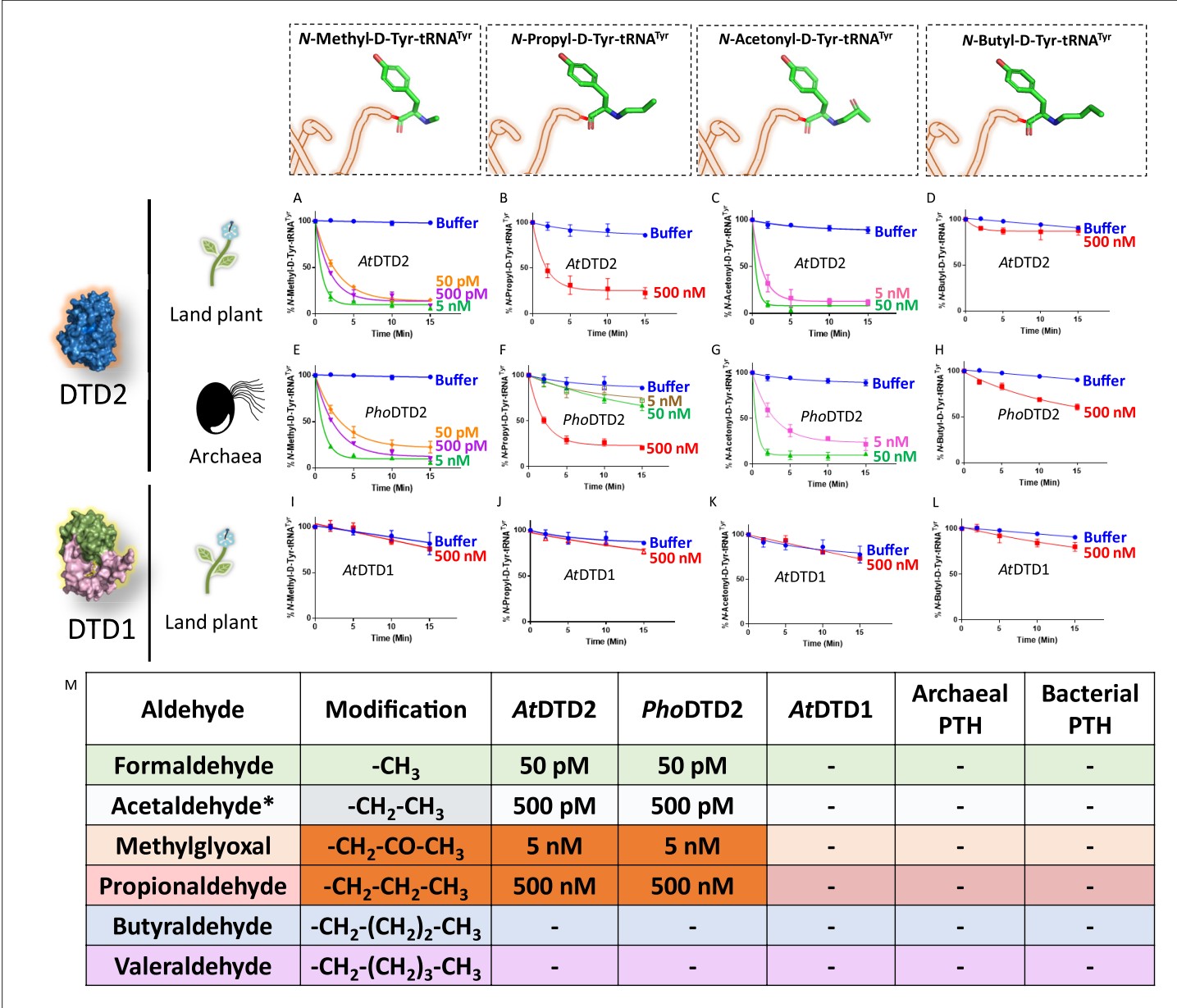

**Figure 3.** D-aminoacyl-tRNA deacylase2 (DTD2) acts as a general aldehyde detoxification system. Deacylation assays on formaldehyde-, propionaldehyde-, methylglyoxal-, and butyraldehyde-modified D-Tyr-tRNA$^{Tyr}$ substrates by *At*DTD2 (**A–D**), *Pho*DTD2 (**E–H**), *At*DTD1 (**I–L**) (n=3). (**M**) Table showing the effective activity concentration of *At*DTD2, *Pho*DTD2, *At*DTD1, archaeal peptidyl-tRNA hydrolase (PTH), and bacterial PTH that completely deacylates aldehyde-modified D-Tyr-tRNA$^{Tyr}$ ('-' denotes no activity; *from ***Mazeed et al., 2021***).

The online version of this article includes the following source data and figure supplement(s) for figure 3:

**Source data 1.** Biochemical data for deacylations of formaldehyde-, propionaldehyde-, MG-, and butyraldehyde-modified D-Tyr-tRNA$^{Tyr}$ substrates by D-aminoacyl-tRNA deacylase1 (DTD1) and DTD2.

**Source data 2.** Biochemical data for deacylations of valeraldehyde- and isovaleraldehyde-modified D-Tyr-tRNA$^{Tyr}$ substrates by D-aminoacyl-tRNA deacylase1 (DTD1) and DTD2.

**Source data 3.** Biochemical data for deacylations of formaldehyde-, propionaldehyde-, MG-, butyraldehyde-, valeraldehyde-, and isovaleraldehyde-modified L-Tyr-tRNA$^{Tyr}$ substrates by D-aminoacyl-tRNA deacylase1 (DTD1), DTD2, peptidyl-tRNA hydrolase1 (PTH1) and PTH2.

**Figure supplement 1.** D-aminoacyl-tRNA deacylase2 (DTD2) is inactive on aldehyde-modified D-aminoacyl-tRNAs (D-aa-tRNAs) beyond three-carbon aldehyde chain length.

**Figure supplement 2.** D-aminoacyl-tRNA deacylase2 (DTD2) acts as a general aldehyde detoxification system.

bacterial EF-Tu (*Thermus thermophilus*) was used. EF-Tu was activated by exchanging the GDP with GTP. Activated EF-Tu protected L-aa-tRNAs from RNase (*Figure 2—figure supplement 1G*). Next, we generated the ternary complex of activated EF-Tu and aa-tRNAs and incubated with formaldehyde. Reaction mixture was quenched at multiple time points and modification was assessed using TLC. It has been seen that activated EF-Tu protected L-aa-tRNAs from smallest aldehyde suggesting that EF-Tu is a dedicated protector of L-aa-tRNAs from all the cellular metabolites (*Figure 2—figure supplement 1H*). However, the lower affinity of D-aa-tRNAs with EF-Tu results in their modification under aldehyde flux. Accumulation of these stable aldehyde-modified D-aa-tRNAs will deplete the free tRNA pool for translation. Therefore, removal of aldehyde-modified D-aa-tRNAs is essential for cell survival.

## DTD2 recycles aldehyde-modified D-aa-tRNAs

Aldehyde-mediated modification on D-aa-tRNAs generated a variety of alkylated-D-aa-tRNA adducts (*Figure 1A* and *Figure 1—figure supplement 1B*). While we earlier showed the ability of DTD2 to remove acetaldehyde-induced modification, we wanted to test whether it can remove diverse range of modifications ranging from smaller methyl to larger valeryl adducts to ensure uninterrupted protein synthesis in plants. To test this, we cloned and purified *Arabidopsis thaliana* (*At*) DTD2 and performed deacylation assays using different aldehyde-modified D-Tyr-(*At*)tRNA$^{Tyr}$ as substrates. DTD2 cleaved majority of aldehyde-modified D-aa-tRNAs at 50 pM to 500 nM range (*Figure 3A–D*, *Figure 3—figure supplement 1A–B* and *Figure 3—figure supplement 2A–F*). Interestingly, DTD2's activity decreases with increase in aldehyde chain lengths (*Figure 3A–D*, *Figure 3—figure supplement 1A–B*, and *Figure 3—figure supplement 2A–F*). To establish DTD2's activity on various aldehyde-modified D-aa-tRNAs as a universal phenomenon, we checked DTD2 activity from an archaeon (*Pyrococcus horikoshii* [*Pho*]). DTD2 from archaea recycled short chain aldehyde-modified D-aa-tRNA adducts as expected (*Figure 3E–G*) and, like DTD2 from plants, it did not act on aldehyde-modified D-aa-tRNAs longer than three carbons (*Figure 3H*, *Figure 3—figure supplement 1C–D*, and *Figure 3—figure supplement 2G–L*). Whereas the canonical chiral proofreader, DTD1, from plants was inactive on all aldehyde-modified D-aa-tRNAs (*Figure 3I–L* and *Figure 3—figure supplement 1E–F*). Interestingly, DTD2 was inactive on butyraldehyde, and higher chain length aldehyde-modified D-aa-tRNAs (*Figure 3D and H*, *Figure 3—figure supplement 1A–D*, *Figure 3—figure supplement 2D–F*, and *Figure 3—figure supplement 2J–L*). This suggests that DTD2 exerts its protection till propionaldehyde with a significant preference for methylglyoxal and formaldehyde-modified D-aa-tRNAs. It is worth noting that the physiological levels of higher chain length aldehydes are comparatively much lesser in plants and archaea (*Figure 1—figure supplement 1A*), indicating the coevolution of DTD2 activity with the presence of toxic aldehydes. Even though both MG and propionaldehyde generate a three-carbon chain modification, DTD2 showed ~100-fold higher activity on MG-modified D-aa-tRNAs (*Figure 3B–C, F–G, and M*). It is interesting to note that peptidyl-tRNA hydrolase (PTH), which recycles on N-acetyl/peptidyl-L-aa-tRNAs and has a similar fold to DTD2, was inactive on formaldehyde and MG-modified L- and D-aa-tRNAs (*Figure 3M* and *Figure 3—figure supplement 2M–Q*). Overall, our biochemical assays with multiple *trans* acting proofreaders (DTD1 and DTD2) and peptidyl-tRNA recycling enzymes (both bacterial and archaeal PTH) suggest that DTD2 is the only aldehyde detoxifier recycling the tRNA pool in both plants and archaea.

## Absence of DTD2 renders plants susceptible to physiologically abundant toxic aldehydes

Biochemical assays suggest that DTD2 may exert its protection for both formaldehyde and MG in addition to acetaldehyde. To test this in vivo, we utilised an *A. thaliana* T-DNA insertion line (SAIL_288_B09) having T-DNA in the first exon of DTD2 gene (*Figure 4A*). We generated a homozygous line (*Figure 4A*) and checked them for ethanol sensitivity as ethanol metabolism produces acetaldehyde. Similar to earlier results (*Fujishige et al., 2004*; *Hirayama et al., 2004*; *Wydau et al., 2007*), dtd2-/- (*dtd2* hereafter) plants were susceptible to ethanol (*Figure 4—figure supplement 1A*) confirming the non-functionality of *DTD2* gene in *dtd2* plants. We then subjected them to various concentrations of formaldehyde and MG generally used for plant toxicity assays (*Welchen et al., 2016*; *Wienstroer et al., 2012*; *Achkor et al., 2003*). These *dtd2* plants were found to be sensitive to both formaldehyde and MG (*Figure 4B–G*). This sensitivity was alleviated by complementing *dtd2* mutant line

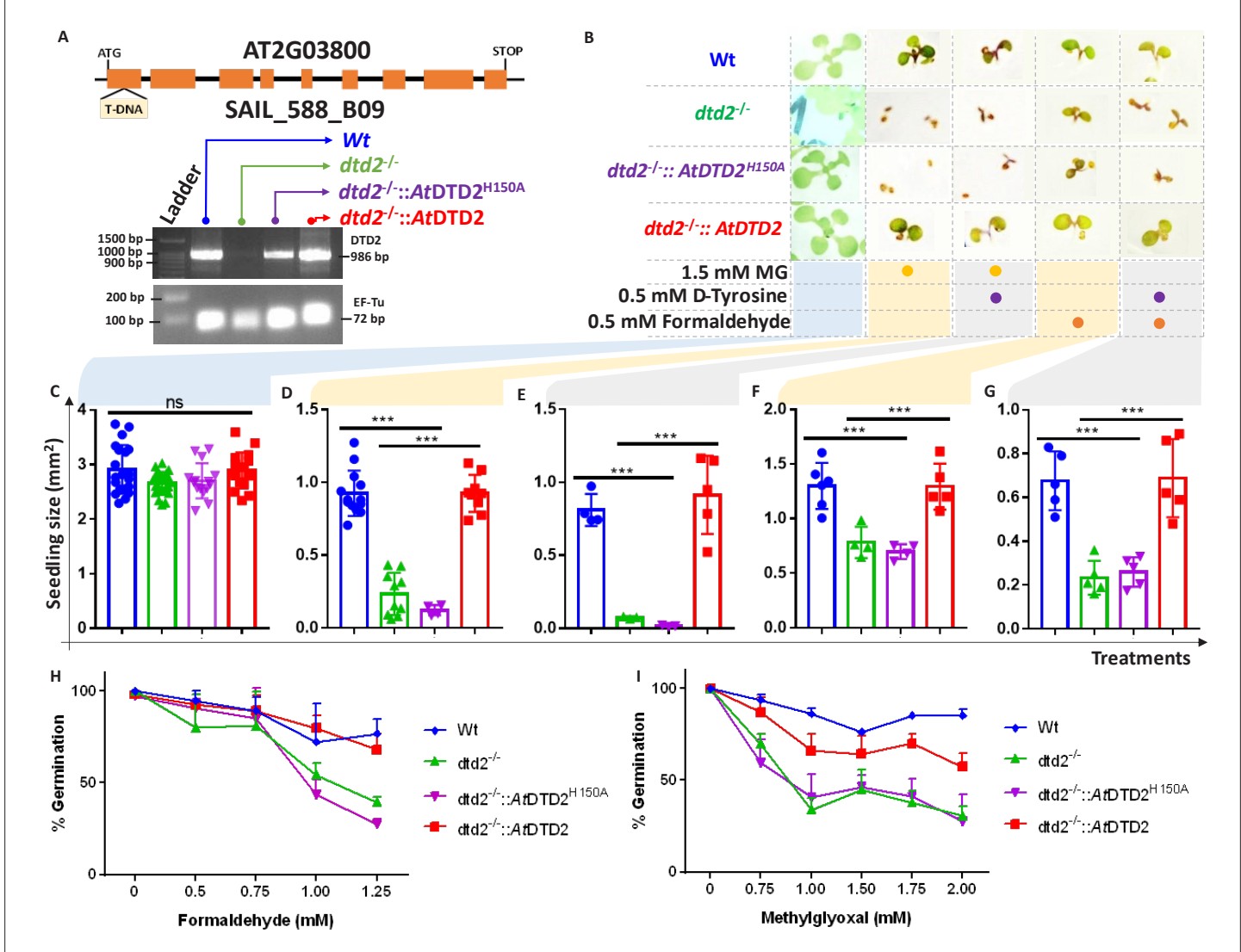

**Figure 4.** D-aminoacyl-tRNA deacylase2 (DTD2) mutant plants are susceptible to physiologically abundant toxic aldehydes. (**A**) Schematics showing the site of T-DNA insertion in (SAIL_288_B09) the first exon of DTD2 gene and reverse transcriptase-polymerase chain reaction (RT-PCR) showing the expression of DTD2 gene in wild type (Wt), *dtd2⁻/⁻*, *dtd2⁻/⁻::At*DTD2 (rescue), and *dtd2⁻/⁻::At*DTD2 H150A (catalytic mutant) plant lines used in the study. (**B**) Toxicity assays showing the effect of formaldehyde and MG with and without D-amino acid (D-tyrosine [D-Tyr]) on *dtd2⁻/⁻* plants. Graph showing the effect of (**C**) Murashige and Skoog agar (MSA), (**D**) 1.5 mM MG, (**E**) 0.5 mM D-Tyr and 1.5 mM MG, (**F**) 0.5 mM formaldehyde, and (**G**) 0.5 mM D-Tyr and 0.5 mM formaldehyde on growth of Wt (Blue), *dtd2⁻/⁻* (Green), *dtd2⁻/⁻::At*DTD2 H150A (catalytic mutant) (purple), and *dtd2⁻/⁻::At*DTD2 (rescue) (red) plants. Cotyledon surface area (mm²) is plotted as parameter for seedling size (n=4–15). Ordinary one-way ANOVA test was used where p values higher than 0.05 are denoted as ns and p≤0.001 are denoted as ***. Graph showing the effect of (**H**) formaldehyde and (**I**) MG on germination of Wt, *dtd2⁻/⁻*, *dtd2⁻/⁻::At*DTD2 (rescue), and *dtd2⁻/⁻::At*DTD2 H150A (catalytic mutant) plants (n=3).

The online version of this article includes the following source data and figure supplement(s) for figure 4:

**Source data 1.** Seedling surface area data and germination data for wild type (Wt), *dtd2⁻/⁻*, *dtd2⁻/⁻::At*DTD2 (rescue), and *dtd2⁻/⁻::At*DTD2 H150A (catalytic mutant) plants under formaldehyde and MG with and without D-amino acid treatments.

**Source data 2.** Germination data for wild type (Wt), *dtd2⁻/⁻*, and *dtd2⁻/⁻::At*DTD2 (rescue) plants under 1.5 mM MG with 0.5 mM D-tyrosine treatment.

**Source data 3.** Biochemical data for deacylations of acetaldehyde-modified D-Tyr-tRNAᵀʸʳ and L-Tyr-tRNAᵀʸʳ substrates by both wild type and catalytic mutant D-aminoacyl-tRNA deacylase2 (DTD2) proteins.

**Source data 4.** Original files for reverse transcriptase-polymerase chain reaction (RT-PCR) analysis in *Figure 4A*.

**Figure supplement 1.** MG and formaldehyde inhibit the germination of D-aminoacyl-tRNA deacylase2 (DTD2) mutant plants.

**Figure supplement 2.** Loss of D-aminoacyl-tRNA deacylase (DTD) results in accumulation of modified D-aminoacyl adducts on tRNAs in *E. coli*.

with genomic copy of wild type DTD2 (*Figure 4A–G*), indicating that DTD2-mediated detoxification plays an important role in plant aldehyde stress. To further confirm the significance of DTD2 in plant growth and development, we performed seed germination assays in *dtd2* plants by evaluating the emergence of radicle on third day post seed plating. As expected, *dtd2* plants show a significant reduction (~40%) in germination (*Figure 4H–I*) and this effect was reversed in the DTD2 rescue line (*Figure 4H–I*). Interestingly, these toxic effects (on both growth and germination) of formaldehyde and MG were enhanced upon D-amino acid supplementation (*Figure 4—figure supplement 1B*). These observations suggest that DTD2's chiral proofreading activity is associated with aldehyde stress removal activity as well. Moreover, to rule out the plausible role of any interacting partner or any other indirect role of DTD2, we generated a catalytic mutant transgenic line containing a genomic copy of *At*DTD2 having H150A mutation (*Ferri-Fioni et al., 2006*; *Figure 4—figure supplement 1C–F*). The catalytic mutant line showed a similar phenotype as *dtd2* plants under aldehyde stress (*Figure 4A–I*), confirming the role of DTD2's biochemical activity in relieving general aldehyde toxicity in plants. We tried to characterise the aldehyde-modified D-aminoacyl adducts on tRNAs with *dtd2* mutant plants extensively through northern blotting as well as mass spectrometry. However, due to the lack of information about the tissue getting affected (root, shoot, etc.), identity of aa-tRNA, as well as location of aa-tRNA (cytosol or organellar), we are so far unsuccessful in identifying them from plants. However, we have used a bacterial surrogate system, *Escherichia coli*, as used earlier (*Mazeed et al., 2021*) to show the accumulation of D-aa-tRNA adducts in the absence of DTD protein. We could identify the accumulation of both formaldehyde- and MG-modified D-aa-tRNA adducts via mass spectrometry (*Figure 4—figure supplement 2A–H*). Overall, our results show that DTD2-mediated detoxification protects plants from physiologically abundant toxic aldehydes.

## Overexpression of DTD2 provides enhanced multi-aldehyde stress tolerance to plants

Plants being sessile are constantly subjected to multiple environmental stresses that reduce agriculture yield and constitute a serious danger to global food security (*Zhu, 2016*). Pyruvate decarboxylase (PDC) transgenics are used to increase flood tolerance in plants but it produces ~35-fold higher acetaldehyde than wild type plants (*Bucher et al., 1994*). Transgenics overexpressing enzymes known for aldehyde detoxification like alcohol dehydrogenase (ADH), aldehyde dehydrogenase (ALDH), aldehyde oxidase (AOX), and glyoxalase are shown to be multi-stress tolerant (*Gupta et al., 2018*; *Zhao et al., 2017*; *Nurbekova et al., 2021*; *Rodrigues et al., 2006*). The sensitivity of *dtd2* plants under physiological aldehydes and biochemical activity of DTD2 prompted us to check if overexpression of DTD2 can provide multi-aldehyde tolerance. We generated a DTD2 overexpression line with DTD2 cDNA cloned under a strong CaMV 35S promoter (*Odell et al., 1985*). We subjected the overexpression line, along with the wild type, to various aldehydes with or without D-amino acids. Strikingly, we found that the DTD2 overexpression line was more tolerant to both the aldehydes (formaldehyde and MG) when compared with wild type (*Figure 5A–C* and *Figure 5—figure supplement 1A–C*). DTD2 overexpression resulted in >50% increased seedling growth when compared with that of wild type (*Figure 5A* and *Figure 5—figure supplement 1D*). The growth difference was more pronounced when D-amino acids were supplemented with varying concentrations of aldehydes (*Figure 5A* and *Figure 5—figure supplement 1D*). Interestingly, DTD2 overexpression plants showed extensive root growth under the influence of both formaldehyde and MG (*Figure 5—figure supplement 1A–C*). Plants produce these aldehydes in huge amounts under various stress conditions (*Jardine et al., 2009*; *Yadav et al., 2005*) and plant tolerance to various abiotic stresses is strongly influenced by root growth (*Seo et al., 2020*). The enhanced root growth by DTD2 overexpression under aldehyde stress implies that DTD2 overexpression offers a viable method to generate multi-stress-resistant crop varieties.

## DTD2 appearance corroborates with the aldehyde burst in land plant ancestors

After establishing the role of DTD2 as a general aldehyde detoxification system in the model land plant system, we wondered if the multi-aldehyde detoxification potential of DTD2 was present in land plant ancestors as well. Therefore, we checked the biochemical activity of DTD2 from a charophyte algae, *Klebsormidium nitens* (*Kn*), and found that it also recycled aldehyde-modified D-aa-tRNAs

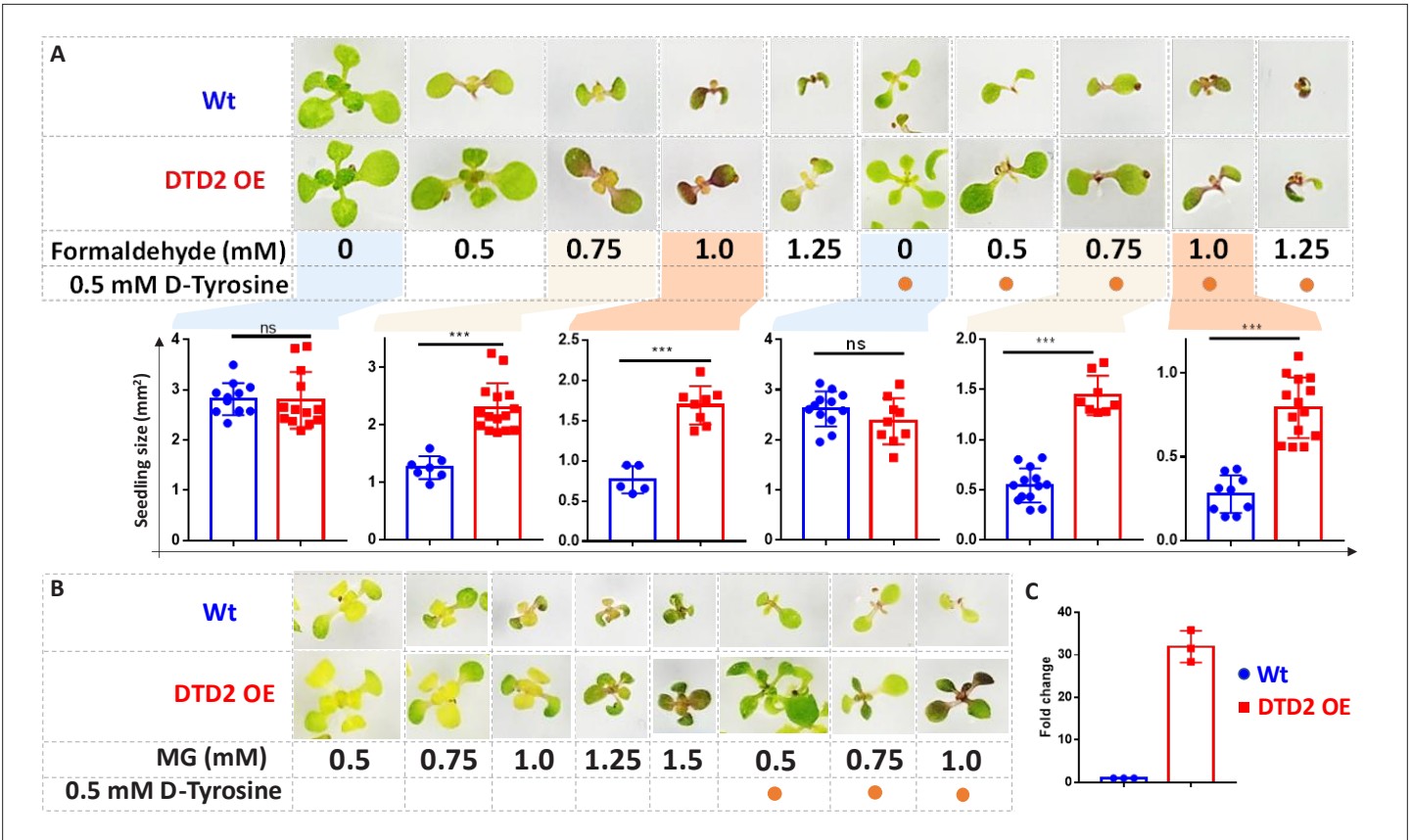

**Figure 5.** Overexpression of D-aminoacyl-tRNA deacylase2 (DTD2) confers increased multi-aldehyde tolerance to *A. thaliana*. DTD2 overexpression (OE) plants grow better than wild type Col-0 under (**A**) 0.5 mM, 0.75 mM, 1.0 mM, and 1.25 mM of formaldehyde with and without 0.5 mM D-tyrosine. Cotyledon surface area (mm²) is plotted as parameter for seedling size (n=5–15). Ordinary one-way ANOVA test was used where p values higher than 0.05 are denoted as ns and p≤0.001 are denoted as ***. (**B**) Growth of DTD2 OE and wild type Col-0 under 0.5 mM, 0.75 mM, 1.0 mM, 1.25 mM, 1.5 mM of MG and 0.5 mM, 0.75 mM, 1.0 mM MG with 0.5 mM D-tyrosine. (**C**) The quantitative polymerase chain reaction (qPCR) analysis showing fold change of DTD2 gene expression in DTD2 OE plant line used (n=3).

The online version of this article includes the following source data and figure supplement(s) for figure 5:

**Source data 1.** Seedling surface area data for wild type (Wt) and D-aminoacyl-tRNA deacylase2 overexpression (DTD2 OE) plants under multiple concentrations of formaldehyde with and without D-amino acid treatments.

**Source data 2.** The quantitative polymerase chain reaction (qPCR) analysis data of D-aminoacyl-tRNA deacylase2 (DTD2) gene expression in wild type (Wt) and DTD2 overexpression (OE) plant line used.

**Source data 3.** Seedling surface area data for wild type (Wt) and D-aminoacyl-tRNA deacylase2 overexpression (DTD2 OE) plants under multiple concentrations of formaldehyde with and without D-amino acid treatments.

**Figure supplement 1.** Overexpression of D-aminoacyl-tRNA deacylase2 (DTD2) confers multi-aldehyde tolerance with D-amino acid stress in *A. thaliana*.

adducts like other plant and archaeal DTD2s (*Figure 6A–C* and *Figure 6—figure supplement 1A–C*). This suggests that the multi-aldehyde problem in plants has its roots in their distant ancestors, charophytes. Next, we analysed the presence of other aldehyde metabolising enzymes across plants. Multiple bioinformatic analyses have shown that land plants encode greater number of ALDH genes compared to green algae (*Tola et al., 2020*; *Islam and Ghosh, 2022*) and glyoxalase family (GlyI, GlyII, and GlyIII), known to clear MG, has expanded exclusively in streptophytic plants (*Singla-Pareek et al., 2020*; *Xu et al., 2023*). We identified that land plants also encode greater number of AOX genes in addition to ALDH genes compared to green algae (*Figure 6—figure supplement 1D*). We delved deeper into plant metabolism with an emphasis on formaldehyde and MG. A search for the formaldehyde (C00067) and MG (C00546) in KEGG database (*Kanehisa et al., 2016*) has shown that formaldehyde is involved in 5 pathways, 60 enzymes, and 94 KEGG reactions, while MG in 6 pathways, 16

enzymes, and 16 KEGG reactions. We did a thorough bioinformatic search for the presence of around 31 and 9 enzymes related to formaldehyde and MG, respectively, in KEGG database (*Supplementary file 1*). Strikingly, we found that plants encode majority of the genes related for formaldehyde and MG and they are conserved throughout land plants (*Figure 6D* and *Figure 6—figure supplement 1E–G*) (*Supplementary file 1*). Plants produce significant amounts of formaldehyde while reshuffling pectin in their cell wall during cell division, development, and tissue damage (*Wu et al., 2018*; *Dorokhov et al., 2018*). Plants contain ~33% pectin in their cell walls that provides strength and flexibility (*Jarvis et al., 1988*). When checked for the presence of genes responsible for the pectin biosynthesis and degradation, we identified that it is a land plant-specific adaptation that originated in early diverging streptophytic algae (*Figure 6—figure supplement 1F*) (*Supplementary file 1*). Overall, our bioinformatic analysis in addition to earlier studies has identified an expansion of aldehyde metabolising repertoire in land plants and their ancestors indicating the sudden aldehyde burst accompanying terrestrialisation which strongly correlates with the recruitment of DTD2 (*Figure 6E* and *Figure 7*).

## Discussion

Plants produce more than 200,000 metabolites for crosstalk with other organisms (*Kessler and Kalske, 2018*). The burgeoning information on increased utilisation of aldehydes for signalling, defence, and altering the ecological interactions with other organisms suggests their physiological importance in plant life (*Yadav et al., 2005*). However, aldehydes are strong electrophiles that undergo addition reactions with amines and thiol groups to form toxic adducts with biomolecules. Excessive aldehyde accumulation irreversibly modifies nucleic acids and proteins resulting in cell death (*Carlsson et al., 2014*; *Pontel et al., 2015*). In this work, we have shown that multiple aldehydes can cause toxicity in *dtd2* plants. Therefore, plants have recruited DTD2 as a detoxifier of aldehyde-induced toxicities in the context of protein biosynthesis. Through this work, we find a correlation between physiological abundance of various aldehydes, their modification propensity, and DTD2's aldehyde protection range. Aldehydes with higher reactivity (formaldehyde, acetaldehyde, and MG) are present in higher amounts in plants and archaea and DTD2 provides modified-D-aa-tRNA deacylase activity against these aldehydes. DTD2's biochemical activity decreases with increase in the aldehyde chain length. Intriguingly, despite MG and propionaldehyde generating a three-carbon long modification, DTD2 is ~100-fold more active on MG-modified D-aa-tRNAs. The absence of carbonyl carbon in the propionaldehyde-modified substrate and DTD2's preferential activity on the bulkier MG-modified substrate points to a clear evolutionary selection pressure for the abundant and physiologically relevant aldehyde. In total contrast to DTD2, all PTH substrates contain carbonyl carbon at the alpha position after the amino group of amino acid in L-aa-tRNA (*Atherly, 1978*). The inactivity of PTH on MG-modified L- and D-aa-tRNAs suggests its specificity for carbonyl carbon at alpha position (*Figure 3—figure supplement 2Q*). Therefore, elucidating the structural basis for both enantioselection and modification specificity of DTD2 and PTH will throw light on these key mechanisms during translation quality control.

The sensitivity of *dtd2* plants to aldehydes of higher prevalence and hyper-propensity for modification indicates the physiological coevolution of aldehyde phytochemistry and recruitment of DTD2 in land plants. Despite the toxic effects of reactive aldehydes, plants are being used as air purifiers as they act as aldehyde scavengers from the environment (*Teiri et al., 2018*; *Aydogan and Montoya, 2011*; *Wang et al., 2014*; *Li et al., 2016*). Moreover, plants have higher removal rates for formaldehyde and acetaldehyde as compared to other higher chain length aldehydes from the environment (*Li et al., 2016*). These aldehydes are produced under various biotic and abiotic stresses in plants and overexpression of enzymes (PDC, ADH, ALDH, and glyoxalase) involved in aldehyde detoxification are shown to provide multi-stress tolerance (*Gupta et al., 2018*; *Quimio et al., 2000*; *Su et al., 2020*; *Sun, 2019*). In similar lines, here, we have also explored the possibility of DTD2 overexpression in multi-aldehyde stress tolerance. Our in vivo results strongly suggests that DTD2 provides multi-aldehyde stress tolerance in the context of detoxifying adducts formed on aa-tRNA. This facilitates the release of free tRNA pool thus relieving translation arrest. DTD2 overexpression plants showed extensive root growth as compared to wild type plants. Plant root growth is an indicator of multi-stress tolerance (*Seo et al., 2020*). Therefore, our DTD2 overexpression approach could be explored further in crop improvement strategies.

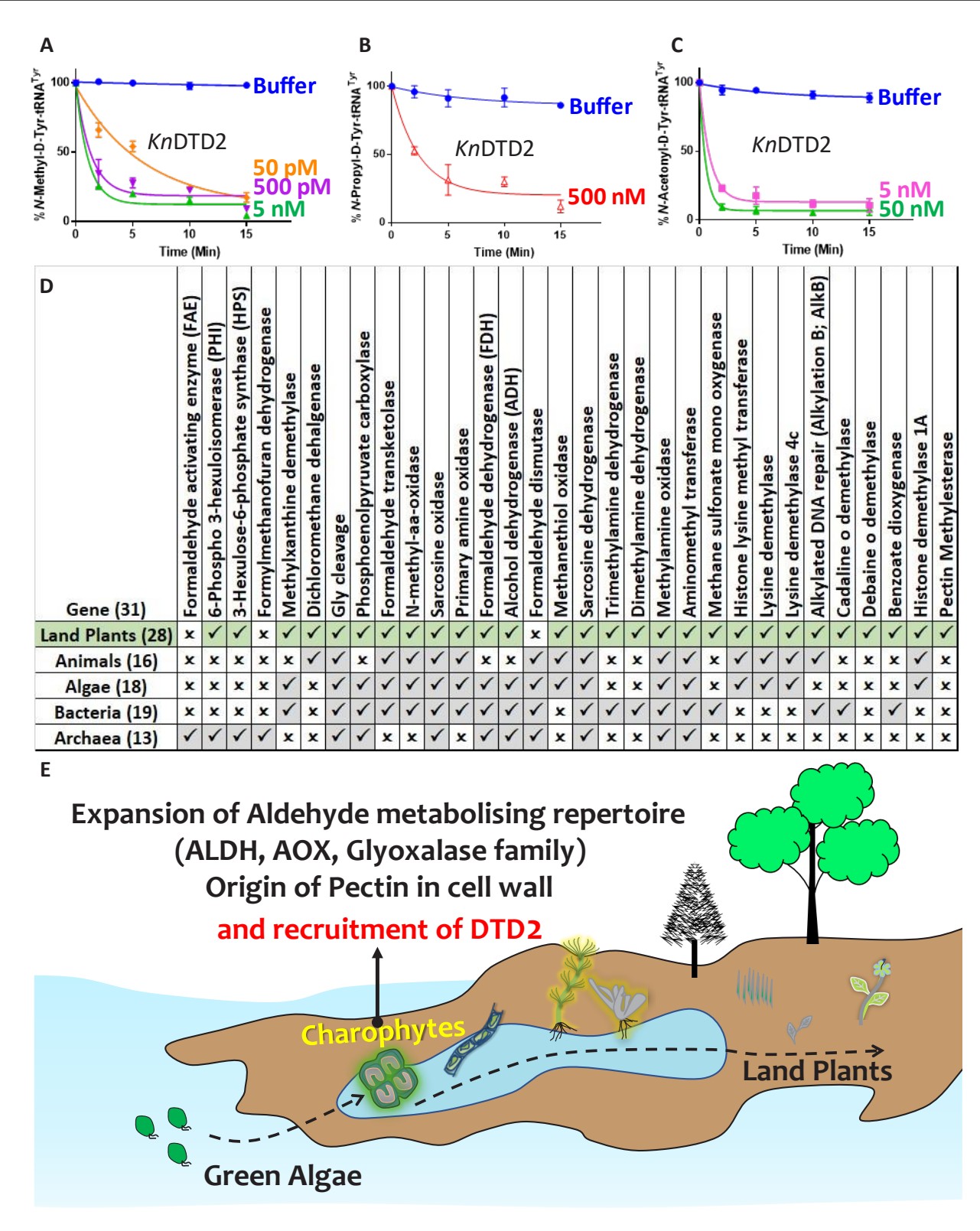

**Figure 6.** Terrestrialisation of plants is associated with expansion of aldehyde metabolising genes. Deacylation assays of *Kn*DTD2 on (**A**) formaldehyde-, (**B**) propionaldehyde-, and (**C**) MG-modified D-Tyr-tRNA<sup>Tyr</sup> (n=3). (**D**) Table showing the presence of 31 genes associated with formaldehyde metabolism in all KEGG organisms across life forms. (**E**) Model showing the expansion of aldehyde metabolising repertoire, cell wall components, and recruitment of archaeal DTD2 in charophytes during land plant evolution.

*Figure 6 continued on next page*

*Figure 6 continued*

The online version of this article includes the following source data and figure supplement(s) for figure 6:

**Source data 1.** Biochemical data for deacylations of formaldehyde-, propionaldehyde-, and MG-modified D-Tyr-tRNA$^{Tyr}$ substrates by KnDTD2.

**Source data 2.** Biochemical data for deacylations of butyraldehyde-, valeraldehyde-, and isovaleraldehyde-modified D-Tyr-tRNA$^{Tyr}$ substrates by KnDTD2.

**Figure supplement 1.** Land plant evolution is associated with the expansion of aldehyde metabolising repertoire.

The role of reactive aldehydes like formaldehyde in the origin of life is inevitable (*Kitadai and Maruyama, 2018*). The presence of reactive aldehydes (*Miller and Urey, 1959*; *Miller, 1957*) and D-amino acids (*Parker et al., 2011*; *Naraoka et al., 2023*) for such a long time suggests an ancient origin of DTD2 activity in last archaeal common ancestor. As archaea thrive in extreme conditions, they secrete enormous amount of formaldehyde into the environment as they grow (*Moran et al., 2016*). We have shown that DTD2 from archaea can efficiently recycle physiologically abundant toxic aldehyde-modified D-aa-tRNAs like plant DTD2s. The adduct removal activity was utilised by the archaeal domain as they produce more aldehydes and thrive in harsh environments (*Gribaldo and Brochier-Armanet, 2006*; *Merino et al., 2019*; *Spang et al., 2017*) and it was later acquired by plants. Bog ecosystems, earlier proposed site for DTD2 gene transfer (*Mazeed et al., 2021*), are highly anaerobic, rich in D-amino acids and ammonia (*Taffner et al., 2018*; *Vranova et al., 2012*; *Kharanzhevskaya et al., 2011*), which lead to enhanced production of aldehydes (acetaldehyde [*Tadege and Kuhlemeier, 1997*] and MG *Borysiuk et al., 2018*) in their inhabitants. Our bioinformatic analysis in addition to earlier studies (*Tola et al., 2020*; *Islam and Ghosh, 2022*; *Singla-Pareek et al., 2020*; *Xu et al., 2023*) has identified an expansion of aldehyde metabolising repertoire exclusively in land plants and their ancestors indicating a sudden aldehyde burst associated with terrestrialisation. Thus, recruitment of archaeal DTD2 by a land plant ancestor must have aided in the terrestrialisation of early land plants. Considering the fact that there are no common incidences of archaeal gene transfer to eukaryotes, it is unclear whether the DTD2 gene was transferred directly to land plant ancestor from archaea or perhaps was mediated by an unidentified intermediate bacterium warrants further investigation. Overall, the study has established the role of archaeal origin DTD2 in land plants by mitigating the toxicity induced by aldehydes during protein biosynthesis.

## Materials and methods

### Plant material and growth conditions

*Arabidopsis* seeds of Columbia background were procured from the Arabidopsis Biological Resource Center (Col-0: CS28166; *dtd2*: SAIL_588_B09 [CS825029]). Plants were cultivated in a growth room at 22°C with 16 hr of light. Seeds were germinated on 1× Murashige-Skoog (MS) medium plates containing 4.4 g/l MS salts, 20 g/l sucrose, and 8 g/l tissue culture agar with pH 5.75 adjusted with KOH at 22°C in a lighted incubator. *Supplementary file 2* contains the primers used to genotype the plants via polymerase chain reaction (PCR).

### Construction of DTD2 rescue and DTD2 overexpression line

The coding sequence for Arabidopsis DTD2 (At2g03800) was PCR-amplified and inserted into pENTR/D-TOPO for the overexpression line and genomic sequence for DTD2 (At2g03800) along with its promoter (~2.4 kb upstream region of DTD2 gene) was PCR-amplified and inserted into pENTR/D--TOPO for the rescue line (primer sequences available in *Supplementary file 2*). Site-directed muta-genesis approach was used to create H150A (catalytic mutant) in plasmid used for rescue line. LR Clonase II (Thermo Fisher Scientific) was used to recombine entry plasmids into (a) pH7FWG2 to create the p35S::DTD2 line and (b) pZP222 to create rescue and catalytic mutant line. *Agrobacterium tumefaciens* Agl1 was transformed with the above destination plasmids. The floral dip technique was then used to transform *Arabidopsis* plants with a Columbia background (*Clough and Bent, 1998*). The transgenic plants for overexpression were selected with 50 µg/ml hygromycin and 1 µg/ml Basta (glufosinate ammonium) and rescue plant lines with 120 µg/ml gentamycin and 1 µg/ml Basta (glufos-inate ammonium) supplemented with MS media.

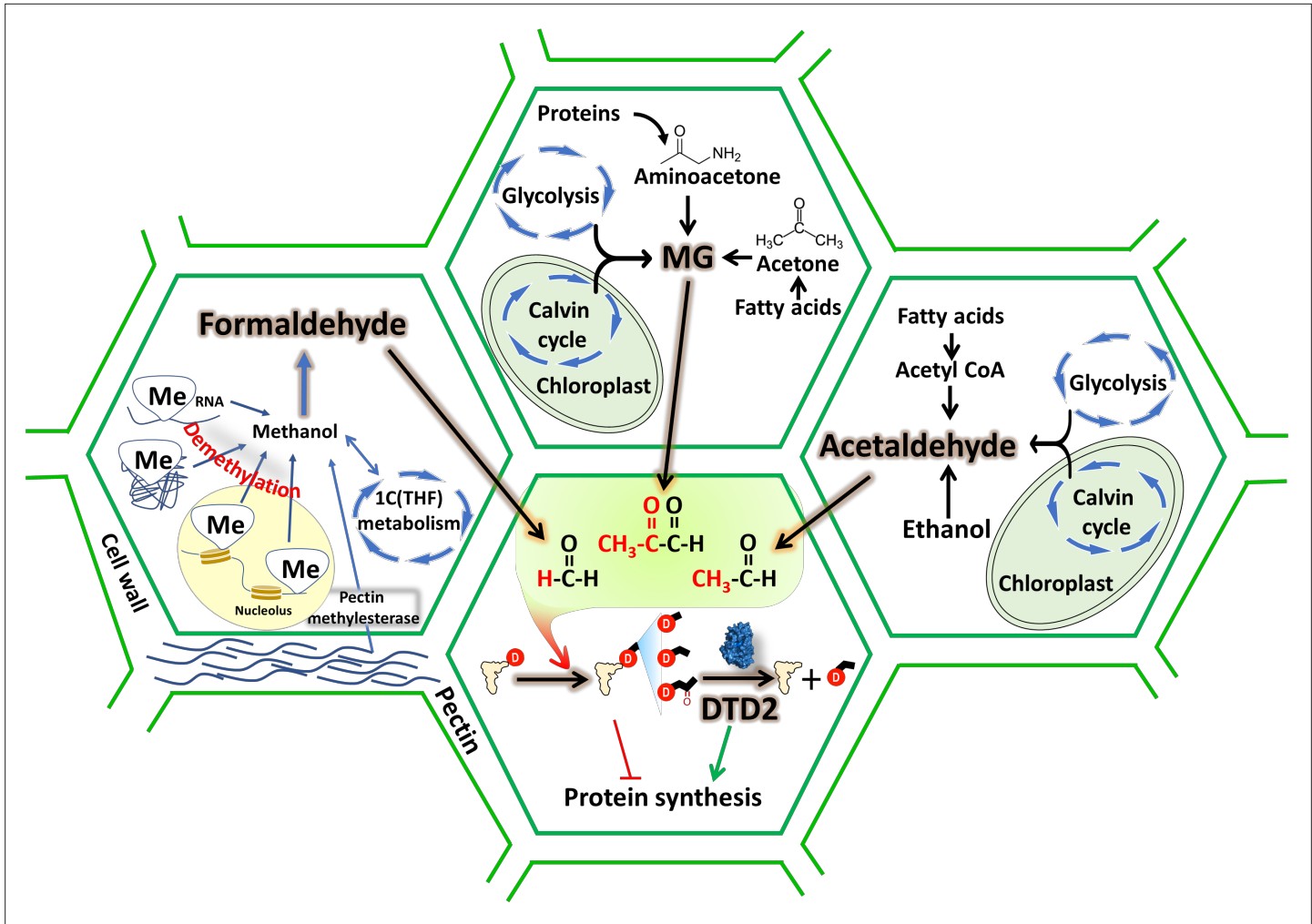

**Figure 7.** D-aminoacyl-tRNA deacylase2 (DTD2) acts as a general aldehyde detoxifier in land plants during translation quality control. Model showing the production of multiple aldehydes like formaldehyde, acetaldehyde, and methylglyoxal (MG) through various metabolic processes in plants. These aldehydes generate stable alkyl modification on D-aminoacyl-tRNA adducts and DTD2 is unique proofreader for these alkyl adducts. Therefore, DTD2 protects plants from aldehyde toxicity associated with translation apparatus emerged from expanded metabolic pathways and D-amino acids.

### Aldehyde sensitivity assays and seedling size quantification

For aldehyde sensitivity assays, seeds were initially sterilised with sterilisation solution and plated on 1× MS medium agar plates containing varying concentrations of aldehydes with or without D-tyrosine. Seeds were grown in a growth room at 22°C with 16 hr of light. Plates were regularly observed and germination percentage was calculated based on the emergence of radicle on third day post seed plating. Phenotypes were documented 2 weeks post germination and seedling size (n=4–15) was quantified. For seedling size quantification imaging was done using Axiozoom stereo microscope with ZEN 3.2 (blue edition) software and processed as necessary. Ordinary one-way ANOVA test was used where p values higher than 0.05 are denoted as ns and p≤0.001 are denoted as ***.

### Total RNA extraction and RT-qPCR

For the reverse transcriptase-quantitative polymerase chain reaction (RT-qPCR) experiment, seeds were germinated and grown for 14 days on MS plate and 200 mg of seedlings were flash-frozen in liquid nitrogen. The RNeasy Plant Minikit (QIAGEN) was used to extract total RNA according to the manufacturer's instructions. 4 µg of total RNA was used for cDNA synthesis with PrimeScript first strand cDNA Synthesis Kit (Takara), according to the manufacturer's instructions. The resultant cDNA was diluted and used as a template for the RT-PCRs for DTD2 rescue and catalytic mutant lines with EF-Tu (At1g07920) as the internal control. While qPCR was done to quantify the level of DTD2

overexpression for DTD2 overexpression line with appropriate primers (*Supplementary file 2*) and Power SYBR Green PCR Master Mix (Thermo Fisher). Reactions were carried out in a Bio-Rad CFX384 thermocycler, with three technical replicates per reaction. The 2-ΔCq method was used for relative mRNA levels calculation with actin (At2g37620) as the internal control. Prism 8 was used for graph generation and statistical analysis.

### Cloning, expression, and purification

DTD1 and DTD2 genes from *A. thaliana* (*At*) were PCR-amplified from cDNA, and DTD2 gene from *K. nitens* (*Kn*) was custom synthesised, while DTD2 gene from *P. horikoshii* (*Pho*) and tyrosyl-tRNA synthetase (TyrRS) of *T. thermophilus* (*Tth*) were PCR-amplified using their genomic DNA with primers listed in *Supplementary file 2*. All the above-mentioned genes were then cloned into the pET28b vector via restriction-free cloning (*van den Ent and Löwe, 2006*). *E. coli* BL21(DE3) was used to overexpress all the above cloned genes except *Ec*PheRS where *E. coli* M15 was used. As plant DTD2s, TyrRS, and PheRS contained 6X His-tag, they were purified via Ni-NTA affinity chromatography, followed by size exclusion chromatography (SEC) using a Superdex 75 column (GE Healthcare Life Sciences, USA). Cation exchange chromatography was used to purify *Pho*DTD2 no-tag protein followed by SEC. Purification method and buffers for all the purifications were used as described earlier (*Ahmad et al., 2013*). All the purified proteins were stored in buffer containing 100 mM Tris (pH 8.0), 200 mM NaCl, 5 mM 2-mercaptoethanol (β-ME), and 50% glycerol for further use.

### Generation of α-$^{32}$P-labelled aa-tRNAs

We have used *A. thaliana* (*At*) tRNA$^{Phe}$, *A. thaliana* (*At*) tRNA$^{Tyr}$, and *E. coli* (*Ec*) tRNA$^{Ala}$ in this study. All the tRNAs were in vitro transcribed using the MEGAshortscript T7 Transcription Kit (Thermo Fisher Scientific, USA). tRNAs were then radiolabelled with [α-$^{32}$P] ATP (BRIT-Jonaki, India) at 3'-end using *E. coli* CCA-adding enzyme (*Ledoux and Uhlenbeck, 2008*). Aminoacylation of tRNA$^{Phe}$, tRNA$^{Tyr}$, and tRNA$^{Ala}$ with phenylalanine, tyrosine, and alanine respectively, were carried out as mentioned earlier (*Ahmad et al., 2013*; *Kuncha et al., 2018*). TLC was used to quantify the aminoacylation as explained (*Mazeed et al., 2021*).

### Generation of adducts on aa-tRNAs for probing relative modification propensity of aldehyde with aa-tRNA and substrate generation for biochemical activity

A single-step method was used for probing relative modification propensity of the aldehyde with aa-tRNA where 0.2 µM of Ala-tRNA$^{Ala}$ was incubated with different concentrations of aldehydes (2 mM and 10 mM) along with 20 mM NaCNBH$_3$ (in 100 mM potassium acetate [pH 5.4]) as a reducing agent at 37°C for 30 min. The reaction mixture was digested with S1 nuclease and analysed on TLC. Except for decanal, all the aldehydes modified Ala-tRNA$^{Ala}$. The method for processing and quantification of modification on aa-tRNA utilised is discussed earlier (*Mazeed et al., 2021*). However, a two-step method was used for generating substrates for biochemical assays as discussed earlier (*Mazeed et al., 2021*). It was used to generate maximum homogenous modification on the aa-tRNAs for deacylation assays. Briefly, 2 µM aa-tRNAs were incubated with 20 mM of formaldehyde, and methylglyoxal or 1 M of propionaldehyde, butyraldehyde, valeraldehyde, and isolvaleraldehyde at 37°C for 30 min. Samples were dried to evaporate excess aldehydes using Eppendorf 5305 Vacufuge plus Concentrator. The dried mixture was then reduced with 20 mM NaCNBH$_3$ at 37°C for 30 min. All reactions were ethanol-precipitated at –30°C overnight or –80°C for 2 hr. Ethanol precipitated pellets were resuspended in 5 mM sodium acetate (pH 5.4) and used for biochemical assays.

### Deacylation assays

For biochemical activity assays, various enzymes like DTD1s, DTD2s, and PTHs were incubated with different aldehyde modified and unmodified α-$^{32}$P-labelled D-Tyr-tRNA$^{Tyr}$ substrates (0.2 µM) in deacylation buffer (20 mM Tris pH 7.2, 5 mM MgCl$_2$, 5 mM DTT, and 0.2 mg/ml bovine serum albumin) at 37°C. An aliquot of 1 µl of the reaction mixture was withdrawn at various time points and digested with S1 nuclease prior to their quantification by TLC. The quantity of aldehyde-modified Tyr-AMP at t=0 min was considered as 100% and the the amount of modified Tyr-AMP at each time point normalised with respect to t=0 min was plotted. All biochemical experiments were repeated at least

three times. The mean values of three independent observations were used to plot the graphs with each error bar representing the standard deviation from the mean value.

## Alkali treatment

Both aldehyde-modified and unmodified D-aa-tRNAs were digested with S1 nuclease before subjecting to alkali treatment (for formaldehyde: 100 nM S1-digested sample with 100 mM Tris pH 9.0; for methylglyoxal: 100 nM S1-digested sample with 200 mM Tris pH 9.0) at 37°C. Alkali-treated samples withdrawn at different time points were directly analysed with TLC. GraphPad Prism software was used to calculate the half-life by fitting the data points onto the curve based on the first-order exponential decay equation $[S_t] = [S_0]e^{-kt}$, where the substrate concentration at time t is denoted as $[S_t]$, $[S_0]$ is the concentration of the substrate at time 0, and k is the first-order decay constant.

## Mass spectrometry

To identify the modification by various aldehydes on D-aa-tRNAs, modified and unmodified D-Phe-tRNA$^{Phe}$ were digested with aqueous ammonia (25% of vol/vol $NH_4OH$) at 70°C for 18 hr (*Mazeed et al., 2021*). Hydrolysed samples were dried using Eppendorf 5305 Vacufuge plus Concentrator. Dried samples were resuspended in 10% methanol and 1% acetic acid in water and analysed via ESI-based mass spectrometry using a Q-Exactive mass spectrometer (Thermo Scientific) by infusing through heated electrospray ionisation source operating at a positive voltage of 3.5 kV. Targeted selected ion monitoring (t-SIM) was used to acquire the mass spectra (at a resolving power of 70,000@200 m/z) with an isolation window of 2 m/z, i.e., theoretical m/z and MH+ ion species. The high energy collision-induced MS-MS spectra with a normalised collision energy of 25 of the selected precursor ion species specified in the inclusion list (having the observed m/z value from the earlier t-SIM analysis) were acquired using the method of t-SIM-ddMS2 (at an isolation window of 1 m/z at a ddMS2 resolving power of 35,000@200 m/z).

## Characterisation of D-aa-tRNA adducts from *E. coli*

To identify the accumulation of D-aa-tRNA adducts, overnight grown primary culture of DTD1 knockout *E. coli* was used to inoculate 1% secondary culture in minimal media with or without 2.5 mM D-tyrosine. Secondary culture grown to $OD_{650}$ (optical density at 650 nm) 0.8 was subjected to respective aldehyde treatment (0.01% final concentration) with 0.5 mM $NaCNBH_3$ at 37°C for 30 min. Cultures were pelleted and total RNA was isolated through acidic phenol chloroform method. Total RNA was digested with three volumes of aqueous ammonia (25% of vol/vol $NH_4OH$) at 70°C for 18 hr (*Mazeed et al., 2021*). Hydrolysed samples were dried using Eppendorf 5305 Vacufuge plus Concentrator. Dried samples were resuspended in 10% methanol and 1% acetic acid in water and analysed via ESI-based mass spectrometry using a Q-Exactive mass spectrometer (Thermo Scientific) as mentioned above.

## Bioinformatic analysis

Protein sequences for various enzymes involved in formaldehyde and MG metabolism were searched in KEGG GENOME database (http://www.genome.jp/kegg/genome.html) (RRID:SCR_012773) through KEGG blast search and all blast hits were mapped on KEGG organisms to identify their taxonomic distribution. KEGG database lacks genome information for charophyte algae so the presence of desired enzymes in charophyte was identified by blast search in NCBI (https://www.ncbi.nlm.nih.gov/) (RRID:SCR_006472). Protein sequences for elongation factor (both EF-Tu and eEF-1a) for the representative organisms were downloaded from NCBI through BLAST-based search. The structure-based multiple sequence alignment of elongation factor was prepared using the T-coffee (http://tcoffee.crg.cat/) (RRID:SCR_011818) server, and the sequence alignment figure was generated using ESPript 3.0 (http://espript.ibcp.fr/ESPript/cgi-bin/ESPript.cgi).

Structure models for elongation factor complexed with aa-tRNA were downloaded from RCSB-PDB (https://www.rcsb.org/) and analysed with The PyMOL Molecular Graphics System, Version 2.0 Schrödinger, LLC. 'ProteinInteractionViewer' plugin for Pymol was used with default parameters to identify and represent the molecular clashes in elongation factor structures with L-phenylalanine and modelled D-phenylalanine, L- and D-alanine in the amino acid binding site of elongation factor. Figures were prepared with The PyMOL Molecular Graphics System, Version 2.0 Schrödinger, LLC.

## Quantification and statistical analysis

Quantification approaches and statistical analyses of the deacylation assays can be found in the relevant sections of the Materials and methods section.

## Acknowledgements

The authors acknowledge Dr. Mukesh Lodha, CSIR-CCMB, for fruitful discussions and Gokulan CG, CSIR-CCMB, for qRT-PCR-related help. PK and SJM thank CSIR, India, for Research Fellowship. RS acknowledges healthcare theme project (MLP-0162, MLP-0138), CSIR, India, JC Bose Fellowship of SERB, India, and Centre of Excellence Project (GAP-0473) of Department of Biotechnology, India.

## Additional information

### Competing interests

Rajan Sankaranarayanan: Reviewing editor, eLife. The other authors declare that no competing interests exist.

### Funding

| Funder | Grant reference number | Author |
| --- | --- | --- |
| Council of Scientific and Industrial Research, India | Research Fellowship | Pradeep Kumar Shivapura Jagadeesha Mukul |
| Council of Scientific and Industrial Research, India | MLP-0138 | Rajan Sankaranarayanan |
| Council of Scientific and Industrial Research, India | MLP-0162 | Rajan Sankaranarayanan |
| Science and Engineering Research Board | J.C. Bose Fellowship | Rajan Sankaranarayanan |
| Department of Biotechnology, Ministry of Science and Technology, India | GAP-0473 | Rajan Sankaranarayanan |

The funders had no role in study design, data collection and interpretation, or the decision to submit the work for publication.

### Author contributions

Pradeep Kumar, Conceptualization, Data curation, Formal analysis, Validation, Investigation, Visualization, Methodology, Writing – original draft, Writing – review and editing; Ankit Roy, Data curation, Formal analysis, Investigation, Visualization, Methodology, Writing – review and editing; Shivapura Jagadeesha Mukul, Investigation, Visualization, Methodology, Writing – review and editing; Avinash Kumar Singh, Resources, Investigation, Methodology, Writing – review and editing; Dipesh Kumar Singh, Resources, Methodology, Writing – review and editing; Aswan Nalli, Resources, Investigation, Writing – review and editing; Pujaita Banerjee, Kandhalu Sagadevan Dinesh Babu, Investigation, Writing – review and editing; Bakthisaran Raman, Investigation, Methodology, Writing – review and editing; Shobha P Kruparani, Imran Siddiqi, Resources, Data curation, Methodology, Writing – review and editing; Rajan Sankaranarayanan, Conceptualization, Resources, Data curation, Formal analysis, Supervision, Funding acquisition, Investigation, Visualization, Methodology, Writing – original draft, Project administration, Writing – review and editing

### Author ORCIDs

Pradeep Kumar ⓘ https://orcid.org/0000-0001-8335-5816
Ankit Roy ⓘ http://orcid.org/0000-0003-3202-9230
Avinash Kumar Singh ⓘ http://orcid.org/0000-0001-6572-6732
Kandhalu Sagadevan Dinesh Babu ⓘ http://orcid.org/0000-0002-1491-1433

Shobha P Kruparani ![ORCID] https://orcid.org/0000-0002-8955-1647
Rajan Sankaranarayanan ![ORCID] https://orcid.org/0000-0003-4524-9953

Reviewer #1 (Public Review): https://doi.org/10.7554/eLife.92827.3.sa1
Reviewer #2 (Public Review): https://doi.org/10.7554/eLife.92827.3.sa2
Author Response https://doi.org/10.7554/eLife.92827.3.sa3

## Additional files

### Supplementary files
• Supplementary file 1. Presence of enzymes related to formaldehyde, MG, and pectin in all KEGG organisms.
• Supplementary file 2. List of DNA primers used in the study.
• MDAR checklist

### Data availability
All data generated or analysed during this study are included in the manuscript and supporting files; source data files have been provided for *Figures 1, 3–6* and associated figure supplements.

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
