## [Editor Report · eLife assessment]

The work is a **fundamental** contribution towards understanding the role of archaeal and plant D-aminoacyl-tRNA deacylase 2 (DTD2) in deacylation and detoxification of D-Tyr-tRNATyr modified by various aldehydes produced as metabolic byproducts in plants. It integrates **convincing** results from both in vitro and in vivo experiments to address the long-standing puzzle of why plants outperform bacteria in handling reactive aldehydes and suggests a new strategy for stress-tolerant crops. A limitation of the study is the lack of evidence for accumulation of toxic D-aminoacyl tRNAs and impairment of translation in plant cells lacking DTD2.

---

## [Referee Report · Reviewer #1 (Public Review)]

Summary: This work is an extension of their earlier work published in Sci Adv in 2021, wherein they showed that DTD2 deacylates N-ethyl-D-aminoacyl-tRNAs arising from acetaldehyde toxicity. The authors (Kumar et al.) in this study, investigate the role of archaeal/plant DTD2 in the deacylation/detoxification of D-Tyr-tRNATyr modified by multiple other aldehydes and methylglyoxal (produced by plants). Importantly, the authors take their biochemical observations to plants, to show that deletion of DTD2 gene from a model plant (*Arabidopsis thaliana*) makes them sensitive to the aldehyde supplementation in the media especially in the presence of D-Tyr. These conclusions are further supported by the observation that the model plant shows increased tolerance to the aldehyde stress when DTD2 is overproduced from the CaMV 35S promoter. The authors propose a model for the role of DTD2 in the evolution of land plants. Finally, the authors suggest that the transgenic crops carrying DTD2 may offer a strategy for stress-tolerant crop development. Overall, the authors present a convincing story, and the data are supportive of the central theme of the story.

Strengths: Data are novel and they provide a new perspective on the role of DTD2, and propose possible use of the DTD2 lines in crop improvement.

Weaknesses: (a) Data obtained from a single aminoacyl-tRNA (D-Tyr-tRNATyr) have been generalized to imply that what is relevant to this model substrate is true for all other D-aa-tRNAs (term modified aa-tRNAs has been used synonymously with the modified Tyr-tRNATyr). This is not a risk-free extrapolation. For example, the authors see that DTD2 removes modified D-Tyr from tRNATyr in a chain-length dependent manner of the modifier. Why do the authors believe that the length of the amino acid side chain will not matter in the activity of DTD2? (b) While the use of EFTu supports that the ternary complex formation by the elongation factor can resist modifications of L-Tyr-tRNATyr by the aldehydes or other agents, in the context of the present work on the role of DTD2 in plants, one would want to see the data using eEF1alpha. This is particularly relevant because there are likely to be differences in the way EFTu and eEF1alpha may protect aminoacyl-tRNAs (for example see description in the latter half of the article by Wolfson and Knight 2005, FEBS Letters 579, 3467-3472).

Note added after revision: The authors have addressed all my concerns by doing additional experiments and by providing convincing arguments. I am happy to conclude that all my concerns on the weaknesses of the work have been nicely addressed. The already convincing story is now stronger.

---

## [Referee Report · Reviewer #2 (Public Review)]

In bacteria and mammals, metabolically generated aldehydes become toxic at high concentrations because they irreversibly modify the free amino group of various essential biological macromolecules. However, these aldehydes can be present in extremely high amounts in archaea and plants without causing major toxic side effects. This fact suggests that archaea and plants have evolved specialized mechanisms to prevent the harmful effects of aldehyde accumulation.

In this manuscript, the authors show that the plant enzyme DTD2, originating from archaea, functions as a D-aminoacyl-tRNA deacylase. This enzyme effectively removes stable D-aminoacyl adducts from tRNAs, enabling these molecules to be recycled for translation. Furthermore, they demonstrate that DTD2 serves as a broad detoxifier for various aldehydes in vivo, extending its function beyond acetaldehyde, as previously believed. Finally, the authors suggest a potential application of their findings by showing that the absence of DTD2 renders plants more susceptible to reactive aldehydes, while its overexpression provides protection against them.

Overall, this study provides a molecular explanation for the remarkable efficiency of plants in handling reactive aldehydes. However, direct evidence that translation is impaired in plants lacking DTD2 experience is currently lacking. Furthermore, because root morphology of DTD2-overexpressing plants appears to differ from that of WT, a thorough phenotypic analysis of DTD2-overexpressing plants will be essential to accurately assess the potential translational application of this enzyme for engineering stress-tolerant plants.

---

## [Author Response]

The following is the authors’ response to the original reviews.

**eLife assessment**
The work is a useful contribution towards understanding the role of archaeal and plant D-aminoacyl-tRNA deacylase 2 (DTD2) in deacylation and detoxification of D-Tyr-tRNATyr modified by various aldehydes produced as metabolic byproducts in plants. It integrates convincing results from both in vitro and in vivo experiments to address the long-standing puzzle of why plants outperform bacteria in handling reactive aldehydes and suggests a new strategy for stress-tolerant crops. The impact of the paper is limited by the fact that only one modified D-aminoacyl tRNA was examined, in lack of evidence that plant eEF1A mimics EF-Tu in protecting L-aminoacyl tRNAs from modification, and in failure to measure accumulation of toxic D-aminoacyl tRNAs or impairment of translation in plant cells lacking DTD2.

We have now addressed all the drawbacks as follows:

‘only one modified D-aminoacyl tRNA was examined’

We wish to clarify that only D-Leu (Yeast), D-Asp (Bacteria, Yeast), D-Tyr (Bacteria, Cyanobacteria, Yeast) and D-Trp (Bacteria) show toxicity in vivo in the absence of known DTD (Soutourina J. et al., JBC, 2000; Soutourina O. et al., JBC, 2004; Wydau S. et al., JBC, 2009) and D-Tyr-tRNATyr is used as a model substrate to test the DTD activity in the field because of the conserved toxicity of D-Tyr in various organisms. DTD2 has been shown to recycle D-Asp-tRNAAsp and D-Tyr-tRNATyr with the same efficiency both in vitro and in vivo (Wydau S. et al., NAR, 2007) and it also recycles acetaldehyde-modified D-Phe-tRNAPhe and D-Tyr-tRNATyr in vitro as shown in our earlier work (Mazeed M. et al., Science Advances, 2021). We have earlier shown that DTD1, another conserved chiral proofreader across bacteria and eukaryotes, acts via a side chain independent mechanism (Ahmad S. et al., eLife, 2013). To check the biochemical activity of DTD2 on D-Trp-tRNATrp, we have now done the D-Trp, D-Tyr and D-Asp toxicity rescue experiments by expressing the archaeal DTD2 in dtd null *E. coli* cells. We found that DTD2 could rescue the D-Trp toxicity with equal efficiency like D-Tyr and D-Asp (Figure: 1). Considering the action on multiple side chains with different chemistry and size, it can be proposed with reasonable confidence that DTD2 also operates based on a side chain independent manner.

**Author response image 1. sa3fig1:** DTD2 recycles multiple D-aa-tRNAs with different side chain chemistry and size. Growth of wildtype (WT), dtd null strain (∆dtd), and Pyrococcus horikoshii DTD2 (PhoDTD2) complemented ∆dtd strains of *E. coli* K12 cells with 500 µM IPTG along with (A) no D-amino acids, (B) 2.5 mM D-tyrosine, (C) 30 mM D-aspartate and (D) 5 mM D-tryptophan.

‘lack of evidence that plant eEF1A mimics EF-Tu in protecting L-aminoacyl tRNAs from modification’

To understand the role of plant eEF1A in protecting L-aa-tRNAs from aldehyde modification, we have done a thorough sequence and structural analysis. We analysed the aa-tRNA bound elongation factor structure from bacteria (PDB ids: 1TTT) and found that the side chain of amino acid in the amino acid binding site of EF-Tu is projected outside (Figure: 2A; 3A). In addition, the amino group of amino acid is tightly selected by the main chain atoms of elongation factor thereby lacking a space for aldehydes to enter and then modify the L-aa-tRNAs and Gly-tRNAs (Figure: 2B; 3B). Modelling of D-amino acid (D-phenylalanine and smallest chiral amino acid, D-alanine) in the same site shows serious clashes with main chain atoms of EF-Tu, indicating D-chiral rejection during aa-tRNA binding by elongation factor (Figure: 2C-E). Next, we superimposed the tRNA bound mammalian eEF-1A cryoEM structure (PDB id: 5LZS) with bacterial structure to understand the structural differences in terms of tRNA binding and found that elongation factor binds tRNA in a similar way (Figure: 3C-D). Modelling of D-alanine in the amino acid binding site of eEF-1A shows serious clashes with main chain atoms, indicating a general theme of D-chiral rejection during aa-tRNA binding by elongation factor (Figure: 2F; 3E). Structure-based sequence alignment of elongation factor from bacteria, archaea and eukaryotes (both plants and mammals) shows a strict conservation of amino acid binding site (Figure: 2G). This suggests that eEF-1A will mimic EF-Tu in protecting L-aa-tRNAs from reactive aldehydes. Minor differences near the amino acid side chain binding site (as indicated in Wolfson and Knight, FEBS Letters, 2005) might induce the amino acid specific binding differences (Figure: 3F). However, those changes will have no influence when the D-chiral amino acid enters the pocket, as the whole side chain would clash with the active site. We have now included this sequence and structural conservation analysis in our revised manuscript (in text: line no 107-129; Figure: 2 and S2). Overall, our structural analysis suggests a conserved mode of aa-tRNA selection by elongation factor across life forms and therefore, our biochemical results with bacterial elongation factor Tu (EF-Tu) reflect the protective role of elongation factor in general across species.

**Author response image 2. sa3fig2:** Elongation factor enantio-selects L-aa-tRNAs through D-chiral rejection mechanism. (A) Surface representation showing the cocrystal structure of EF-Tu with L-Phe-tRNAPhe. Zoomed-in image showing the binding of L-phenylalanine with side chain projected outside of binding site of EF-Tu (PDB id: 1TTT). (B) Zoomed-in image of amino acid binding site of EF-Tu bound with L-phenylalanine showing the selection of amino group of amino acid through main chain atoms (PDB id: 1TTT). (C) Modelling of D-phenylalanine in the amino acid binding site of EF-Tu shows severe clashes with main chain atoms of EF-Tu. Modelling of smallest chiral amino acid, alanine, in the amino acid binding site of EF-Tu shows (D) no clashes with L-alanine and (E) clashes with D-alanine. (F) Modelling of D-alanine in the amino acid binding site of eEF-1A shows clashes with main chain atoms. (*Represents modelled molecule). (G) Structure-based sequence alignment of elongation factor from bacteria, archaea and eukaryotes (both plants and animals) showing conserved amino acid binding site residues. (Key residues are marked with red star).

**Author response image 3. sa3fig3:** Elongation factor protects L-aa-tRNAs from aldehyde modification. (A) Cartoon representation showing the cocrystal structure of EF-Tu with L-Phe-tRNAPhe (PDB id: 1TTT). (B) Zoomed-in image of amino acid binding site of EF-Tu bound with L-phenylalanine (PDB id: 1TTT). (C) Cartoon representation showing the cryoEM structure of eEF-1A with tRNAPhe (PDB id: 5LZS). (D) Image showing the overlap of EF-Tu:L-Phe-tRNAPhe crystal structure and eEF-1A:tRNAPhe cryoEM structure (r.m.s.d. of 1.44 Å over 292 Cα atoms). (E) Zoomed-in image of amino acid binding site of eEF-1A with modelled L-alanine (PDB id: 5ZLS). (*Modelled) (F) Overlap showing the amino acid binding site residues of EF-Tu and eEF-1A. (EF-Tu residues are marked in black and eEF-1A residues are marked in red).

‘failure to measure accumulation of toxic D-aminoacyl tRNAs or impairment of translation in plant cells lacking DTD2’

We agree that measuring the accumulation of D-aa-tRNA adducts from plant cells lacking DTD2 is important. We tried to characterise the same with dtd2 mutant plants extensively through Northern blotting as well as mass spectrometry. However, due to the lack of information about the tissue getting affected (root or shoot), identity of aa-tRNA as well as location of aa-tRNA (cytosol or organellar), we are so far unsuccessful in identifying them from plants. Efforts are still underway to identify them from plant system lacking DTD2. However, we have used a bacterial surrogate system, *E. coli,* as used earlier in Mazeed M. et al., Science Advances, 2021 to show the accumulation of D-aa-tRNA adducts in the absence of dtd. We could identify the accumulation of both formaldehyde and MG modified D-aa-tRNA adducts via mass spectrometry (Figure: 4). These results are now included in the revised manuscript (in line no: 190-197 and Figure: S5).

**Author response image 4. sa3fig4:** Loss of DTD results in accumulation of modified D-aminoacyl adducts on tRNAs in E. coli. Mass spectrometry analysis showing the accumulation of aldehyde modified D-Tyr-tRNATyr in (A) Δdtd *E. coli*, (B) formaldehyde and D-tyrosine treated Δdtd *E. coli*, and (C) MG and D-tyrosine treated Δdtd *E. coli*. ESI-MS based tandem fragmentation analysis for unmodified and aldehyde modified D-Tyr-tRNATyr in (D) Δdtd *E. coli*, (E) and (F) formaldehyde and D-tyrosine treated Δdtd *E. coli*, (G) and (H) MG and D-tyrosine treated Δdtd *E. coli*.

**Response to Public Reviews:**

We are grateful for the reviewers’ positive feedback and their comments and suggestions on this manuscript. Reviewer 1 has indicated two weaknesses and Reviewer 2 has none. We have now addressed all the concerns of the Reviewers.

**Reviewer #1 (Public Review):**
Summary:This work is an extension of the authors' earlier work published in Sci Adv in 2001, wherein the authors showed that DTD2 deacylates N-ethyl-D-aminoacyl-tRNAs arising from acetaldehyde toxicity. The authors in this study, investigate the role of archaeal/plant DTD2 in the deacylation/detoxification of D-Tyr-tRNATyr modified by multiple other aldehydes and methylglyoxal (produced by plants). Importantly, the authors take their biochemical observations to plants, to show that deletion of DTD2 gene from a model plant (*Arabidopsis thaliana*) makes them sensitive to the aldehyde supplementation in the media especially in the presence of D-Tyr. These conclusions are further supported by the observation that the model plant shows increased tolerance to the aldehyde stress when DTD2 is overproduced from the CaMV 35S promoter. The authors propose a model for the role of DTD2 in the evolution of land plants. Finally, the authors suggest that the transgenic crops carrying DTD2 may offer a strategy for stress-tolerant crop development. Overall, the authors present a convincing story, and the data are supportive of the central theme of the story.

We are happy that reviewer found our work convincing and would like to thank the reviewer for finding our data supportive to the central theme of the manuscript.

Strengths:Data are novel and they provide a new perspective on the role of DTD2, and propose possible use of the DTD2 lines in crop improvement.

We are happy for this positive comment on the manuscript.

Weaknesses:(a) Data obtained from a single aminoacyl-tRNA (D-Tyr-tRNATyr) have been generalized to imply that what is relevant to this model substrate is true for all other D-aa-tRNAs (term modified aa-tRNAs has been used synonymously with the modified Tyr-tRNATyr). This is not a risk-free extrapolation. For example, the authors see that DTD2 removes modified D-Tyr from tRNATyr in a chain-length dependent manner of the modifier. Why do the authors believe that the length of the amino acid side chain will not matter in the activity of DTD2?

We thank the reviewer for bringing up this important point. As mentioned above, we wish to clarify that only half of the aminoacyl-tRNA synthetases are known to charge D-amino acids and only D-Leu (Yeast), D-Asp (Bacteria, Yeast), D-Tyr (Bacteria, Cyanobacteria, Yeast) and D-Trp (Bacteria) show toxicity in vivo in the absence of known DTD (Soutourina J. et al., JBC, 2000; Soutourina O. et al., JBC, 2004; Wydau S. et al., JBC, 2009). D-Tyr-tRNATyr is used as a model substrate to test the DTD activity in the field because of the conserved toxicity of D-Tyr in various organisms. DTD2 has been shown to recycle D-Asp-tRNAAsp and D-Tyr-tRNATyr with the same efficiency both in vitro and in vivo (Wydau S. et al., NAR, 2007). Moreover, we have previously shown that it recycles acetaldehyde-modified D-Phe-tRNAPhe and D-Tyr-tRNATyr in vitro as shown in our earlier work (Mazeed M. et al., Science Advances, 2021). We have earlier shown that DTD1, another conserved chiral proofreader across bacteria and eukaryotes, acts via a side chain independent mechanism (Ahmad S. et al., eLife, 2013). To check the biochemical activity of DTD2 on D-Trp-tRNATrp, we have now done the D-Trp, D-Tyr and D-Asp toxicity rescue experiments by expressing the archaeal DTD2 in dtd null *E. coli* cells. We found that DTD2 could rescue the D-Trp toxicity with equal efficiency like D-Tyr and D-Asp (Figure 1). Considering the action on multiple side chains with different chemistry and size, it can be proposed with reasonable confidence that DTD2 also operates based on a side chain independent manner.

(b) While the use of EFTu supports that the ternary complex formation by the elongation factor can resist modifications of L-Tyr-tRNATyr by the aldehydes or other agents, in the context of the present work on the role of DTD2 in plants, one would want to see the data using eEF1alpha. This is particularly relevant because there are likely to be differences in the way EFTu and eEF1alpha may protect aminoacyl-tRNAs (for example see description in the latter half of the article by Wolfson and Knight 2005, FEBS Letters 579, 3467-3472).

We thank the reviewer for bringing up this important point. As mentioned above, to understand the role of plant eEF1A in protecting L-aa-tRNAs from aldehyde modification, we have done a thorough sequence and structural analysis. We analysed the aa-tRNA bound elongation factor structure from bacteria (PDB ids: 1TTT) and found that the side chain of amino acid in the amino acid binding site of EF-Tu is projected outside (Figure: 2A; 3A). In addition, the amino group of amino acid is tightly selected by the main chain atoms of elongation factor thereby lacking a space for aldehydes to enter and then modify the L-aa-tRNAs and Gly-tRNAs (Figure: 2B; 3B). Modelling of D-amino acid (D-phenylalanine and smallest chiral amino acid, D-alanine) in the same site shows serious clashes with main chain atoms of EF-Tu, indicating D-chiral rejection during aa-tRNA binding by elongation factor (Figure: 2C-E). Next, we superimposed the tRNA bound mammalian eEF-1A cryoEM structure (PDB id: 5LZS) with bacterial structure to understand the structural differences in terms of tRNA binding and found that elongation factor binds tRNA in a similar way (Figure: 3C-D). Modelling of D-alanine in the amino acid binding site of eEF-1A shows serious clashes with main chain atoms, indicating a general theme of D-chiral rejection during aa-tRNA binding by elongation factor (Figure: 2F; 3E). Structure-based sequence alignment of elongation factor from bacteria, archaea and eukaryotes (both plants and mammals) shows a strict conservation of amino acid binding site (Figure: 2G). Minor differences near the amino acid side chain binding site (as indicated in Wolfson and Knight, FEBS Letters, 2005) might induce the amino acid specific binding differences (Figure: 3F). However, those changes will have no influence when the D-chiral amino acid enters the pocket, as the whole side chain would clash with the active site. We have now included this sequence and structural conservation analysis in our revised manuscript (in text: line no 107-129; Figure: 2 and S2). Overall, our structural analysis suggests a conserved mode of aa-tRNA selection by elongation factor across life forms and therefore, our biochemical results with bacterial elongation factor Tu (EF-Tu) reflect the protective role of elongation factor in general across species.

**Reviewer #2 (Public Review):**
In bacteria and mammals, metabolically generated aldehydes become toxic at high concentrations because they irreversibly modify the free amino group of various essential biological macromolecules. However, these aldehydes can be present in extremely high amounts in archaea and plants without causing major toxic side effects. This fact suggests that archaea and plants have evolved specialized mechanisms to prevent the harmful effects of aldehyde accumulation.In this study, the authors show that the plant enzyme DTD2, originating from archaea, functions as a D-aminoacyl-tRNA deacylase. This enzyme effectively removes stable D-aminoacyl adducts from tRNAs, enabling these molecules to be recycled for translation. Furthermore, they demonstrate that DTD2 serves as a broad detoxifier for various aldehydes in vivo, extending its function beyond acetaldehyde, as previously believed. Notably, the absence of DTD2 makes plants more susceptible to reactive aldehydes, while its overexpression offers protection against them. These findings underscore the physiological significance of this enzyme.

We thank the reviewer for the positive comments the manuscript.

**Response to recommendation to authors:**

**Reviewer #1 (Recommendations For The Authors):**
I enjoyed reading the manuscript entitled, "Archaeal origin translation proofreader imparts multi aldehyde stress tolerance to land plants" from the Sankaranarayanan lab. This work is an extension of their earlier work published in Sci Adv in 2001, wherein they showed that DTD2 deacylates N-ethyl-D-aminoacyl-tRNAs arising from acetaldehyde toxicity. Now, the authors of this study (Kumar et al.) investigate the role of archaeal/plant DTD2 in the deacylation/detoxification of D-Tyr-tRNATyr modified by multiple other aldehydes and methylglyoxal (which are produced during metabolic reactions in plants). Importantly, the authors take their biochemical observations to plants, to show that deletion of DTD2 gene from a model plant (*Arabidopsis thaliana*) makes them sensitive to the aldehyde supplementation in the media especially in the presence of D-Tyr. These conclusions are further supported by the observation that the model plant shows increased tolerance to the aldehyde stress when DTD2 is overproduced from the CaMV 35S promoter. The authors propose a model for the role of DTD2 in the evolution of land plants. Finally, the authors suggest that the transgenic crops carrying DTD2 may offer a strategy for stress-tolerant crop development. Overall, the authors present a convincing story, and the data are supportive of the central theme of the story.

We are happy that reviewer enjoyed our manuscript and found our work convincing. We would also like to thank reviewer for finding our data supportive to the central theme of the manuscript.

I have the following observations that require the authors' attention.1. The title of the manuscript will be more appropriate if revised to, "Archaeal origin translation proofreader, DTD2, imparts multialdehyde stress tolerance to land plants".

Both the reviewer’s suggested to change the title. We have now changed the title based on reviewer 2 suggestion.

1. Abstract (line 19): change, "physiologically abundantly produced" to "physiologically produced".

As per the reviewer’s suggestion, we have now changed it to "physiologically produced".

1. Introduction (line 50): delete, 'extremely'.

We have removed the word 'extremely' from the Introduction.

1. Line 79: change, "can be utilized" to "may be explored".

We have changed "can be utilized" to "may be explored" as suggested by the reviewers.

1. Results in general:(a) Data obtained from a single aminoacyl-tRNA (D-Tyr-tRNATyr) have been generalized to imply that what is relevant to this model substrate is true for all other D-aa-tRNAs (term modified aa-tRNAs has been used synonymously with the modified D-Tyr-tRNATyr). This is a risky extrapolation. For example, the authors see that DTD2 removes modified D-Tyr from tRNATyr in a chain-length dependent manner of the modifier. Why do the authors believe that the length of the amino acid side chain will not matter in the activity of DTD2?

We thank the reviewer for bringing up this important point. As mentioned above, we wish to clarify that only half of the aminoacyl-tRNA synthetases are known to charge D-amino acids and only D-Leu (Yeast), D-Asp (Bacteria, Yeast), D-Tyr (Bacteria, Cyanobacteria, Yeast) and D-Trp (Bacteria) show toxicity in vivo in the absence of known DTD (Soutourina J. et al., JBC, 2000; Soutourina O. et al., JBC, 2004; Wydau S. et al., JBC, 2009). D-Tyr-tRNATyr is used as a model substrate to test the DTD activity in the field because of the conserved toxicity of D-Tyr in various organisms. DTD2 has been shown to recycle D-Asp-tRNAAsp and D-Tyr-tRNATyr with the same efficiency both in vitro and in vivo (Wydau S. et al., NAR, 2007). Moreover, we have previously shown that it recycles acetaldehyde-modified D-Phe-tRNAPhe and D-Tyr-tRNATyr in vitro as shown in our earlier work (Mazeed M. et al., Science Advances, 2021). We have earlier shown that DTD1, another conserved chiral proofreader across bacteria and eukaryotes, acts via a side chain independent mechanism (Ahmad S. et al., eLife, 2013). To check the biochemical activity of DTD2 on D-Trp-tRNATrp, we have now done the D-Trp, D-Tyr and D-Asp toxicity rescue experiments by expressing the archaeal DTD2 in dtd null *E. coli* cells. We found that DTD2 could rescue the D-Trp toxicity with equal efficiency like D-Tyr and D-Asp (Figure 1). Considering the action on multiple side chains with different chemistry and size, it can be proposed with reasonable confidence that DTD2 also operates based on a side chain independent manner.

(b) Interestingly, the authors do suggest (in the Materials and Methods section) that the experiments were performed with Phe-tRNAPhe as well as Ala-tRNAAla. If what is stated in Materials and Methods is correct, these data should be included to generalize the observations.

We regret for the confusing statement. We wish to clarify that L- and D-Tyr-tRNATyr were used for checking the TLC-based aldehyde modification, EF-Tu based protection assays and deacylation assays, D-Phe-tRNAPhe was used to characterise aldehyde-based modification by mass spectrometry and L-Ala-tRNAAla was used to check the modification propensity of multiple aldehydes. We used multiple aa-tRNAs to emphasize that aldehyde-based modifications are aspecific towards the identity of aa-tRNAs. All the data obtained with respective aa-tRNAs are included in manuscript.

(c) While the use of EFTu supports that the ternary complex formation by the elongation factor can resist modifications of L-Tyr-tRNATyr by the aldehydes or other agents, in the context of the present work on the role of DTD2 in plants, one would want to see the data using eEF1alpha. This is particularly relevant because there are likely to be differences in the way EFTu and eEF1alpha may protect aminoacyl-tRNAs (for example see description in the latter half of the article by Wolfson and Knight 2005, FEBS Letters 579, 3467-3472).

We thank the reviewer for bringing up this important point. As mentioned above, to understand the role of plant eEF1A in protecting L-aa-tRNAs from aldehyde modification, we have done a thorough sequence and structural analysis. We analysed the aa-tRNA bound elongation factor structure from bacteria (PDB ids: 1TTT) and found that the side chain of amino acid in the amino acid binding site of EF-Tu is projected outside (Figure: 2A; 3A). In addition, the amino group of amino acid is tightly selected by the main chain atoms of elongation factor thereby lacking a space for aldehydes to enter and then modify the L-aa-tRNAs and Gly-tRNAs (Figure: 2B; 3B). Modelling of D-amino acid (D-phenylalanine and smallest chiral amino acid, D-alanine) in the same site shows serious clashes with main chain atoms of EF-Tu, indicating D-chiral rejection during aa-tRNA binding by elongation factor (Figure: 2C-E). Next, we superimposed the tRNA bound mammalian eEF-1A cryoEM structure (PDB id: 5LZS) with bacterial structure to understand the structural differences in terms of tRNA binding and found that elongation factor binds tRNA in a similar way (Figure: 3C-D). Modelling of D-alanine in the amino acid binding site of eEF-1A shows serious clashes with main chain atoms, indicating a general theme of D-chiral rejection during aa-tRNA binding by elongation factor (Figure: 2F; 3E). Structure-based sequence alignment of elongation factor from bacteria, archaea and eukaryotes (both plants and mammals) shows a strict conservation of amino acid binding site (Figure: 2G). Minor differences near the amino acid side chain binding site (as indicated in Wolfson and Knight, FEBS Letters, 2005) might induce the amino acid specific binding differences (Figure: 3F). However, those changes will have no influence when the D-chiral amino acid enters the pocket, as the whole side chain would clash with the active site. We have now included this sequence and structural conservation analysis in our revised manuscript (in text: line no 107-129; Figure: 2 and S2). Overall, our structural analysis suggests a conserved mode of aa-tRNA selection by elongation factor across life forms and therefore, our biochemical results with bacterial elongation factor Tu (EF-Tu) reflect the protective role of elongation factor in general across species.

1. Results (line 89): Figure: 1C-G (not B-G).

As correctly pointed out by the reviewer(s), we have changed it to Figure: 1C-G.

1. Results (line 91): Figure: S1B-G (not C-G).

We wish to clarify that this is correct.

1. Line 97: change, "propionaldehyde" to "propionaldehyde (Figure: 1H)".

As per the reviewer’s suggestion, we have now changed, "propionaldehyde" to "propionaldehyde (Figure: 1H)".

1. Line 124: The statement, "DTD2 cleaved all modified D-aa-tRNAs at 50 pM to 500 nM range (Figure: 2A_D)" is not consistent with the data presented. For example, Figure 2D does not show any significant cleavage. Figure S2A-B also does not show cleavage.

We thank the reviewers for pointing this out. We have changed the sentence to “DTD2 cleaved majority of aldehyde modified D-aa-tRNAs at 50 pM to 500 nM range".

1. Line 131: Cleavage observed in Fig. S2E is inconsistent with the generalized statement on DTD1.

We wish to clarify that the minimal activity seen in Fig. S2E is inconsistent with the general trend of DTD1’s biochemical activity seen on modified D-aa-tRNAs. In addition, we have earlier shown that D-aa-tRNA fits snugly in the active site of DTD1 (Ahmad S. et al., eLife, 2013) whereas the modified D-aa-tRNA cannot bind due to the space constrains in the active site of DTD1 (Mazeed M. et al., Science Advances, 2021). Therefore, this minimal activity could be a result of technical error during this biochemical experiment and could be considered as no activity.

1. Lines 129-133: Citations of many figure panels particularly in the supplementary figures are inconsistent with generalized statements. This section requires a major rewrite or rearrangement of the figure panels (in case the statements are correct).

We thank the reviewers for bringing forth this point and we have accordingly modified the statement into “DTD2 from archaea recycled short chain aldehyde-modified D-aa-tRNA adducts as expected (Figure: 3E-G) and, like DTD2 from plants, it did not act on aldehyde-modified D-aa-tRNAs longer than three chains (Figure: 3H; S3C-D; S4G-L)”.

1. Line 142: I don't believe one can call PTH a proofreader. Its job is to recycle tRNAs from peptidyl-tRNAs.

We thank the reviewers for pointing out this very important point. This is now corrected.

13). Line 145: change, "DTD2 can exert its protection for" to "DTD2 may exert protection from".

As per the reviewer’s suggestion, we have now changed"DTD2 can exert its protection for" to "DTD2 may exert protection from".

1. Line 148: change, "a homozygous line (Figure: 3A) and checked for" to "homozygous lines (Figure: 3A) and checked them for".

As per the reviewer’s suggestion, we have now changed, "a homozygous line (Figure: 3A) and checked for" to "homozygous lines (Figure: 3A) and checked them for".

1. Line 148: Change, the sentence beginning with dtd2 as follows. Similar to earlier results30-32, dtd2-/- (dtd2 hereafter) plants were susceptible to ethanol (Figure: S4A) confirming the non-functionality DTD2 gene in dtd2 plants.

As per the reviewer’s suggestion, we have now changed the sentence accordingly.

1. Line 161: change, "linked" to "associated".

As per the reviewer’s suggestion, we have now changed "linked" to "associated".

1. Lines 173-176: It would be interesting to know how well the DTD2 OE lines do in comparison to the other known transgenic lines developed with, for example, ADH, ALDH, or AOX lines. Any ideas would help appreciate the observation with DTD2 OE lines!

We greatly appreciate the reviewer’s suggestion. We have not done any comparison experiment with any transgenic lines so far. However, it can be potentially done in further studies with DTD2 OE lines.

1. Line 194: change, "necessary" with "present".

As per the reviewer’s suggestion, we have now changed "necessary" with "present".

1. Line 210: what is meant by 'huge'? Would 'significant' sound better?

As per the reviewer’s suggestion, we have now changed "huge" with "significant".

1. Lines 239-243: This needs to be rephrased. Isn't alpha carbonyl of the carboxyl group that makes ester bond with the -CCA end of the tRNA required for DTD2 activity as well? Are you referring to the carbonyl group in the moiety that modifies the alpha-amino group? Please clarify. The cited reference (no. 64) of Atherly does not talk about it.

We regret for the confusing statement. To clarify, we were referencing to the carbonyl carbon of the modification post amino group of the amino acid in aa-tRNAs (Figure: 5). We have now included a figure (Figure: S4Q of revised manuscript) to show the comparison of the carbonyl group for the better clarity. The cited reference Atherly A. G., Nature, 1978 shows the activity of PTH on peptidyl-tRNAs and peptidyl-tRNAs possess carbonyl carbon at alpha position post amino group of amino acid in L-aa-tRNAs.

**Author response image 5. sa3fig5:** Figure showing the difference in the position of carbonyl carbon in acetonyl and acetyl modification on aa-tRNAs.

1. Line 261: thrive (not thrives).

As per the reviewer’s suggestion, we have now changed it to thrive.

1. In Fig3A: second last lane, it should be dtd-/-:: AtDTDH150A (not dtd-/-:: AtDTDH150A).

We thank the reviewers for pointing out this, we have corrected it.

23). Materials and methods: Please clarify which experiments used tRNAPhe, tRNAAla, PheRS, etc. Also, please carefully check all other details provided in this section.

As per the reviewer’s suggestion, we would like to provide a table below explaining the use of different substrates as well as enzymes in our experiments.

**Author response table 1. sa3table1:** 

Component	Purpose
tRNA^(Tyr)	To check the aldehyde modification, EF-Tu based protection assays and check theenzymatic activity of various enzymes like DTD1, DTD2 and PTH
tRNAPhe	To characterise the chemical nature of the modification via mass spectrometry
tRNA Ala	To check the modification propensity of aldehydes with varying chain length
TyrRS	To generate L-Tyr-tRNA Tyr and D-Tyr-tRNA Tyr for biochemical assays
PheRS	To generate D-Phe-tRNAPhe for mass-spectrometry-based assays
AlaRS	To generate L-Ala-tRNA Ala for checking the modification strength of differentaldehydes

1. Figure legends (many places): p values higher than 0.05 (not less than) are denoted as ns.

We thank the reviewers for pointing out this. We have corrected it.

**Reviewer #2 (Recommendations For The Authors):**
I have only minor comments for the authors:Title: I would replace "Archeal origin translation proofreader" with " A translation proofreader of archeal origin"

As per the reviewer’s suggestion, we have now changed the title.

Abstract: This section could benefit from some rewriting. For instance, at the outset, the initial logical connection between the first and second sentences of the abstract is somewhat unclear. At the very least, I would suggest swapping their order to enhance the narrative flow. Later in the text, the term "chiral proofreading systems" is introduced; however, it is only in a subsequent sentence that these systems are explained to be responsible for removing stable D-aminoacyl adducts from tRNA. Providing an immediate explanation of these systems would enhance the reader's comprehension. The authors switch from the past participle tense to the present tense towards the end of the text. I would recommend that they choose one tense for consistency. In the final sentence, I would suggest toning down the statement and replacing "can be used" with "could be explored." (https://www.nature.com/articles/d41586-023-02895-w). The same comment applies to the introduction, line 79.

As per the reviewer’s suggestion, we have now changed the abstract appropriately.

General note: Conventionally, the use of italics is reserved for the specific species "*Arabidopsis thaliana*," while the broader genus "Arabidopsis" is not italicized.

We acknowledge the reviewer for this pertinent suggestion. This is now corrected in revised version of our manuscript.

General note: I would advise the authors against employing bold characters in conjunction with colors in the figures.

We thank the reviewer for this suggestion. We have now changed it appropriately in revised version of our manuscript.

Figure 1A: I recommend including the concentrations of the various aldehydes used in the experiment within the figure legend. While this information is available in the materials and methods section, it would be beneficial to have it readily accessible when analyzing the figure.

As per the reviewer’s suggestion, we have now included the concentrations in figure legend.

Figure 1I, J: some error bars are invisible.

We thank the reviewers for pointing out this, we have corrected it.

Figure 2M: The table could be simplified by removing aldehydes for which it was not feasible to demonstrate activity. The letter "M" within the cell labeled "aldehydes" appears to be a typographical error, presumably indicating the figure panel.

As per the reviewer’s suggestion, we have now changed this appropriately.

Figure 3: For consistency with the other panels in the figure, I recommend including an additional panel to display the graph depicting the impact of MG on germination.

As per the reviewer’s suggestion, we have now changed this appropriately.

Figure 4: Considering that only one plant is presented, it would be beneficial to visualize the data distribution for the other plants used in this experiment, similar to what the authors have done in panel A of the same figure.

We thank the reviewer for bringing up this point. We wish to clarify that we have done experiment with multiple plants. However, for the sake of clarity, we have included the representative images. Moreover, we have included the quantitative data for multiple plants in Figure 3C-G.

Figure 5E: The authors may consider presenting a chronological order of events as they believe they occurred during evolution.

We thank the reviewer for the suggestion. However, it is very difficult to pinpoint the chronology of the events. Aldehydes are lethal for systems due to their hyper reactivity and systems would require immediate solutions to survive. Therefore, we think that both problem (toxic aldehyde production) and its solution (expansion of aldehyde metabolising repertoire and recruitment of archaeal DTD2) might have appeared simultaneously.

Figure 6: The model appears somewhat crowded, which may affect its clarity and ease of interpretation. The authors might also consider dividing the legend sentence into two separate sentences for better readability.

As per the reviewer’s suggestion, we have now changed this appropriately.

Line 149: I recommend explicitly stating that ethanol metabolism produces acetaldehyde. This clarification will help the general reader immediately understand why DTD2 mutant plants are sensitive to ethanol.

As per the reviewer’s suggestion, we have now changed this appropriately.

Line 289: there is a typographical error, "promotor" instead of the correct term "promoter.".

We thank the referee for pointing out this, we have now corrected it.

Figure S5: The root morphology of DTD2 OE plants appears to exhibit some differences compared to the WT, even in the absence of a high concentration of aldehydes. It would be valuable if the authors could comment on these observed differences unless they have already done so, and I may have overlooked it.

We thank the referee for pointing out this. We do see minor differences in root morphology, but they are more pronounced with aldehyde treatments. The reason for this phenotype remains elusive and we are trying to understand the role of DTD2 in root development in detail in further studies.

Some Curiosity Questions (not mandatory for manuscript acceptance):1. Do DTD2 OE plants display an earlier flowering phenotype than wild-type Col-0?

We have not done detailed phenotyping of DTD2 OE plants. However, our preliminary observations suggest no differences in flowering pattern as compared to wild-type Col-0.

1. What is the current understanding of the endogenous regulation of DTD2?

We have not done detailed analysis to understand the endogenous regulation of DTD2.

1. Could the protective phenotype of DTD2 OE plants in the presence of aldehydes be attributed to additional functions of this enzyme beyond the removal of stable D-aminoacyl adducts from tRNAs?

Based on the available evidence regarding the biochemical activity and in vivo phenotypes of DTD2, it appears that removal of stable D-aminoacyl adducts from tRNA is key for the protective phenotype of DTD2 OE.

A Suggestion for Future Research (not required for manuscript acceptance):The authors could explore the possibility of overexpressing DTD2 in pyruvate decarboxylase transgenic plants and assess whether this strategy enhances flood tolerance without incurring a growth penalty under normal growth conditions.

We thank the referee for this interesting suggestion for future research. We will surely keep this in mind while exploring the flood tolerance potential of DTD2 OE plants.